# Drought-exposure history increases complementarity between plant species in response to a subsequent drought

Yuxin Chen [1,2,3✉], Anja Vogel[4,5,6], Cameron Wagg [2,7], Tianyang Xu[2], Maitane Iturrate-Garcia[2,8], Michael Scherer-Lorenzen [9], Alexandra Weigelt [4,10], Nico Eisenhauer [4,5] & Bernhard Schmid [11,12✉]

Growing threats from extreme climatic events and biodiversity loss have raised concerns about their interactive consequences for ecosystem functioning. Evidence suggests biodiversity can buffer ecosystem functioning during such climatic events. However, whether exposure to extreme climatic events will strengthen the biodiversity-dependent buffering effects for future generations remains elusive. We assess such transgenerational effects by exposing experimental grassland communities to eight recurrent summer droughts versus ambient conditions in the field. Seed offspring of 12 species are then subjected to a subsequent drought event in the glasshouse, grown individually, in monocultures or in 2-species mixtures. Comparing productivity between mixtures and monocultures, drought-selected plants show greater between-species complementarity than ambient-selected plants when recovering from the subsequent drought, causing stronger biodiversity effects on productivity and better recovery of drought-selected mixtures after the drought. These findings suggest exposure to recurrent climatic events can improve ecosystem responses to future events through transgenerational reinforcement of species complementarity.

[1] Key Laboratory of the Ministry of Education for Coastal and Wetland Ecosystems, College of the Environment & Ecology, Xiamen University, 361102 Xiamen, China. [2] Department of Evolutionary Biology and Environmental Studies, University of Zürich, Winterthurerstrasse 190, 8057 Zürich, Switzerland. [3] School of Life Sciences/State Key Laboratory of Biocontrol, Sun Yat-sen University, 510275 Guangzhou, China. [4] German Centre for Integrative Biodiversity Research (iDiv) Halle-Jena-Leipzig, Puschstraße 4, 04103 Leipzig, Germany. [5] Institute of Biology, Leipzig University, Puschstraße 4, 04103 Leipzig, Germany. [6] Institute of Ecology and Evolution, Friedrich Schiller University Jena, 07743 Jena, Germany. [7] Fredericton Research and Development Center, Agriculture and Agri-Food Canada, 850 Lincoln Road, Fredericton, New Brunswick E3B 4Z7, Canada. [8] Department of Chemical and Biological Metrology, Federal Institute of Metrology METAS, Lindenweg 50, 3003 Bern-Wabern, Switzerland. [9] Geobotany, Faculty of Biology, University of Freiburg, 79106 Freiburg, Germany. [10] Institute of Biology, Leipzig University, Johannisallee 21-23, 04103 Leipzig, Germany. [11] Department of Geography, Remote Sensing Laboratories, University of Zürich, Winterthurerstrasse 190, 8057 Zürich, Switzerland. [12] Institute of Ecology, College of Urban and Environmental Sciences, Peking University, 100871 Beijing, China. ✉email: yuxin.chen@xmu.edu.cn; bernhard.schmid@uzh.ch

Extreme climatic events such as droughts are predicted to be more frequent in the future[1,2], with potentially negative effects on the functioning of ecosystems and the provision of ecosystem services for human well-being[3–7]. Higher biodiversity may buffer the impacts of a single drought event through increasing the resistance against the drought-driven loss of ecosystem functioning or the recovery of ecosystem functioning after the event[8–12] (but see refs. [13,14]). However, droughts can increase the risks of species loss in species-rich ecosystems[15], which may reduce the sustainability of the stabilizing effects of biodiversity in the face of recurrent droughts. Species complementarity (i.e., less competitive or more facilitative interactions between species than within species) is a crucial mechanism driving both the stabilizing effects of biodiversity and biodiversity maintenance[16,17]. If exposure to droughts of previous generations can increase species complementarity in future generations, the stabilizing effects of biodiversity can be more sustainable in a future with more frequent droughts. However, whether recurrent extreme climatic events can cause transgenerational reinforcement of species complementarity remains unknown.

Transgenerational effects or inclusive inheritance can arise from both genetic and non-genetic transmission of phenotypic variation between generations[18–20]. First, biotic or abiotic environmental changes may lead to rapid evolutionary change, defined as change in gene frequencies in populations, through a filtering of pre-existing genetic variation via differential survival or proliferation of specific genotypes or, less likely, recombination or mutation[21,22]. For example, 11 years of selection by community diversity for perennial plants in a grassland biodiversity experiment caused the rapid emergence of populations with different genetic composition[23]. Second, environmental changes can also cause epigenetic modifications without genetic change, such as DNA methylation, histone modifications, and noncoding RNA expression[19,20]. These transgenerational epigenetic effects may permit more rapid adaptation of organisms to environmental changes than evolutionary mechanisms[24]. Third, transgenerational non-genetic effects can also arise from processes without an epigenetic basis. For example, parent–offspring transmission of phenotypic variation may result from modifications of nutrients or hormonal information of parents[20,25]. Environmental changes may trigger one or all of the above mechanisms[19].

Transgenerational changes in plant traits can modify species interactions and complementarity in different ways. First, shifts of plant trait distributions may be independent of community composition or biodiversity, but vary between genotypes or species. This can restructure the patterns of trait dissimilarity within communities, which would further modify interactions and complementarity between species[26,27]. For example, droughts may select for specific plant traits with the consequence of higher tolerance of water deficits (e.g., shorter height, greater leaf mass per area, higher turgor loss point, or higher root-shoot biomass ratio)[28–30] or faster recovery after droughts (e.g., with lower leaf mass per area, or higher leaf and root nitrogen content)[29,31,32]. The strength of selection for traits that confer higher tolerance or faster recovery may differ between species with conservative and acquisitive resource use strategies. Second, trait selection in one species may depend on traits of other interacting species within the same communities[33]. For example, experiments have shown that microbial[34,35] and plant[36–38] species evolved in more diverse communities had less negative interactions or more complementarity between species under ambient environment, potentially strengthening the positive biodiversity effects on productivity. Whether and how extreme climatic events will lead to transgenerational changes in species interactions and complementarity, and further change the biodiversity effects on ecosystem functioning in plant communities, remains elusive.

In this study, we aim to investigate three questions: (1) how recurrent summer droughts over an 8-year period in a field experiment influence biodiversity effects on productivity and stability over different phases of a subsequent drought event in a glasshouse experiment; (2) whether altered species interactions drive the transgenerational responses to drought that underlie biodiversity effects on productivity and stability; and (3) which traits mediate the above transgenerational changes in biodiversity effects and species interactions. Inspired by the stress gradient hypothesis[39], which predicts that interactions among plants are less competitive or more facilitative in stressful environments where resources are harder to access, we hypothesize that offspring from plants with drought-selection history show less negative interactions and more complementarity between species than those from plants with ambient-selection history, leading to more positive biodiversity effects on ecosystem functioning.

We expose experimental communities of grassland species to eight recurrent summer droughts vs. ambient conditions in a long-term biodiversity experiment in the field (the Jena Experiment[11,40,41]). Evolution by selection from standing genetic variation but not by epigenetic change was indicated in previous studies from the Jena Experiment in the ambient treatment[23,36]. Seed offspring of 12 species are subsequently grown individually, in monocultures, or in 2-species mixtures and subjected to a subsequent drought event in the glasshouse (Fig. 1; Supplementary Table 1). In the glasshouse experiment, both monocultures and mixtures contain four individuals per pot. We harvest aboveground biomass (a proxy for productivity) of all individuals in each pot three times (Fig. 1): (1) after a first phase of 3 months with regular watering (ambient conditions, "before drought"); (2) after a second phase of regrowth under regular watering followed by 2 weeks without watering (drought conditions, "during drought"); and (3) after a third phase of 7 weeks with regular watering for recovery (ambient conditions, "after drought"). After the first harvest, plants are watered regularly and allowed to regrow before the drought event. This harvesting procedure mimics the common cutting management of the species in the field (and in comparable grasslands in the region), where up to four harvests per growing season are being made[42]. To avoid confusion between the drought treatment in the field vs. the one in the glasshouse, we name the former selection treatment. We call plants with drought-selection history drought-selected plants and plants with ambient-selection history ambient-selected plants.

To investigate the three questions proposed above, we use the harvested aboveground biomass in four sets of analyses. First, we assess how drought selection affects biodiversity effects on productivity before, during, and after the drought event in the glasshouse. We define the biomass difference between 2-species mixtures and monocultures as the net biodiversity effect (a positive net biodiversity effect represents a higher observed productivity in a mixture than that expected from corresponding monocultures), which we partition (Eq. (1) in "Methods") into a complementarity effect (a positive complementarity effect arises from niche partitioning or facilitation between species in a mixture) and a sampling effect (a positive sampling effect arises from larger contributions to mixture productivity of species that are more productive in monocultures; which is also termed "selection effect" elsewhere, a term we did not use in this study to avoid confusion with the term "selection treatment")[17]. Second, we assess the effects of drought selection on resistance (biomass ratio during vs. before the drought), recovery (biomass ratio after vs. during the drought), and resilience (biomass ratio after vs. before the drought) of productivity[9,43]. Third, we test whether altered plant interactions drive the above differences between the two

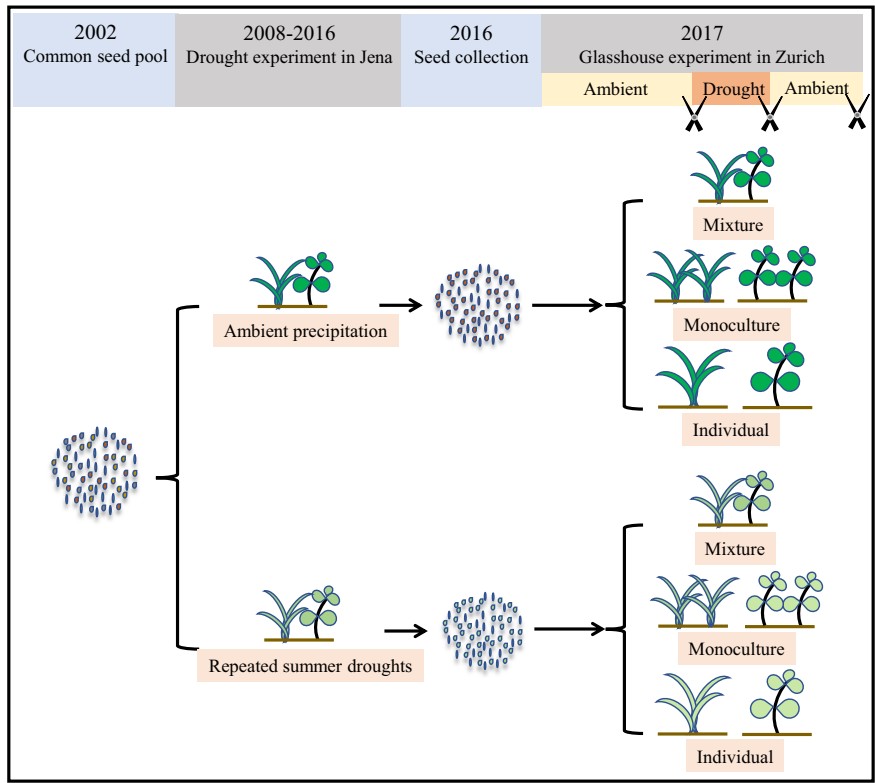

**Fig. 1 Experimental design.** Seeds collected from different selection treatments (8-years treatments of recurrent summer droughts vs. ambient control) in the Jena Experiment, Germany, were sown in 2-species mixtures, in monocultures, or individually in pots in a glasshouse at the University of Zurich, Switzerland. During a first phase of 3 months, pots were watered regularly (ambient conditions, "before drought"). This was followed by a second phase of 2 weeks without watering (drought conditions, "during drought"). Finally, a third phase of 7 weeks with regular watering allowed the plants to recover and regrow after the experimental drought (ambient conditions for recovery, "after drought"). Plants were harvested after 14–16 weeks (before the drought), 20 weeks (at the end of the drought), and 27 weeks (after recovery from the drought) as represented by scissors. After the first harvest, plants were watered regularly and allowed to regrow before the drought event.

selection treatments. We calculate the intensities of neighbor interactions, $NInt_M$, by comparing plant biomass between individual-plant pots and multi-plant pots (Eq. (2) in "Methods")[44]. We conduct the calculation of neighbor interactions separately for mixtures (i.e., heterospecific interactions) and monocultures (i.e., conspecific interactions), and then calculate their difference as a proxy for niche difference. Fourth, we assess how drought selection influences trait values on plants in pots with one individual and trait dissimilarities between interacting species in mixtures.

Here, we show that the 8-year selection treatment of recurrent droughts in the field increases niche differentiation and complementarity between species in mixtures during the recovery phase after an experimental drought event in the glasshouse. This leads to more positive biodiversity effects on community productivity and recovery after the drought event.

### Results

**Biodiversity effects on productivity.** We first tested the biodiversity effects per species pair, separately for each selection treatment and harvest time. We found that the net biodiversity effects on productivity for different species pairs were higher when plants had been selected under repeated summer droughts in the field, but this only became visible when their productivity was assessed after the drought event in the glasshouse (Supplementary Fig. 1). The positive net biodiversity effects in mixtures of drought-selected plants were mostly due to positive complementarity effects (CEs) (Supplementary Fig. 1). We found

significant positive correlations between the CEs of species pairs before and during the drought event in the glasshouse, for both ambient- and drought-selected plants (Supplementary Table 2). However, species pairs of drought-selected plants reversed in their ranks in CEs before vs. after the subsequent drought event (i.e., a negative correlation; Supplementary Fig. 2). This rank reversal was not present for species pairs of ambient-selected plants (Supplementary Fig. 2).

The presence or absence of specific species in drought-selected species pairs did not significantly change CEs after the subsequent drought event (Supplementary Data 1). These results suggest that the increased CEs of drought-selected plants after the subsequent drought event were a general phenomenon and not due to particular species or species pairs with large effects on CEs.

Next, we tested the biodiversity effects across all species pairs and the two selection treatments in combined statistical analyses for each harvest time. Testing the overall effect of drought-selection against the variation among species pairs as an error term, we found that the drought-selection treatment after the subsequent drought event led to significantly increased net biodiversity and complementarity effects and more negative sampling effects across all species pairs tested in the glasshouse experiment, confirming the generality of the individual findings reported above (Fig. 2; Table 1). These results suggest that different drought-selected species are more complementary than ambient-selected species in mixtures when recovering from drought. However, before and during the subsequent drought event, the effects of drought-selection were not statistically

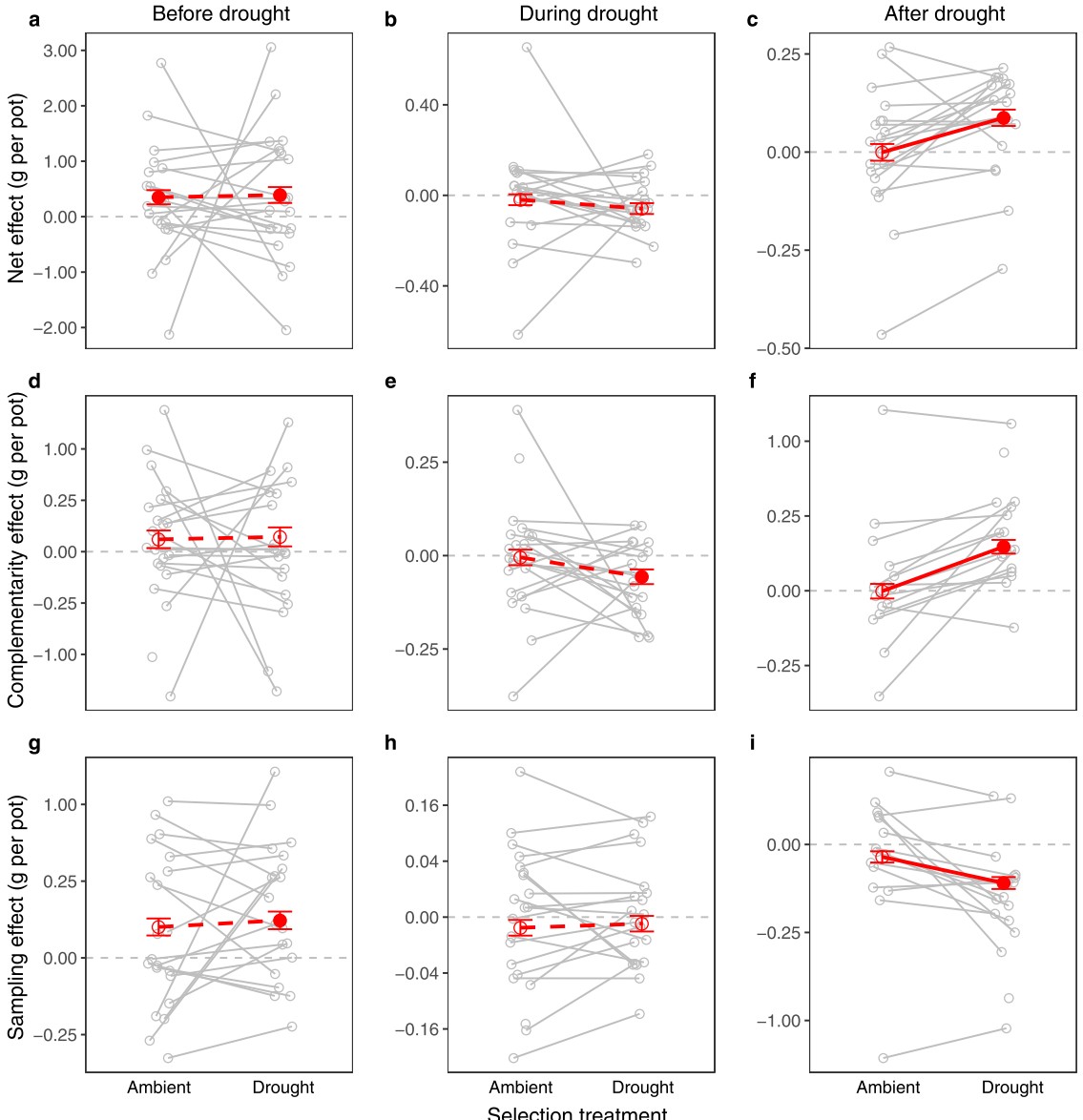

**Fig. 2 Difference in biodiversity effects on productivity between selection treatments (ambient- vs. drought-selected plants) before, during, and after the drought event in the glasshouse.** Difference in biodiversity effects on productivity between selection treatments (ambient- vs. drought-selected plants) before, during, and after the drought event in the glasshouse. Biodiversity effects on productivity were calculated as net effect (**a–c**), complementarity effect (**d–f**), and sampling effect (**g–i**). The solid red lines indicate significant differences between the two selection treatments (see Table 1). Red points and error bars show means ± standard error of 2-species mixtures of the corresponding selection treatment. The filled red points show significant differences from zero (horizontal dashed lines, see Supplementary Table 3), i.e., positive or negative average biodiversity effects on productivity. Gray points represent means for species pairs (standard errors for species pairs not shown). Gray lines connect the same species pair between the two selection treatments. Source data are provided as a Source data file. The numbers of mixtures per species pair (gray points) and per selection treatment (red points) are provided in the Source data file and Supplementary Table 3, respectively.

significant (Fig. 2; Table 1). Before the subsequent drought event, the positive biodiversity effects were mainly due to positive sampling effects (Fig. 2a, d, g).

Although the drought- and ambient-selected plants in our glasshouse experiments came from different diversity levels in the Jena Experiment[11,40,41] (see Supplementary Data 2), we could not find any significant effects of original diversity treatments (originated from monoculture field plots or not; originated from field plots with different functional group richness) nor significant interactions between field diversity treatments and the drought-selection history (Supplementary Data 3–5). Therefore, we excluded the history of biodiversity treatments in the field from further analyses.

**Stability.** Drought-selected plants recovered faster from the subsequent drought event in the glasshouse than did ambient-selected plants; however, this was only evident when plants were grown in mixtures but not in monocultures (Supplementary Fig. 3; Supplementary Table 4). These results suggest that species interactions may play important roles in promoting the recovery of drought-selected plants in mixtures. In contrast to these differences in recovery, plants with different selection treatments did not differ significantly in their resistance or resilience to the subsequent drought event (Supplementary Fig. 3; Supplementary Table 4).

The difference in recovery rates between mixtures and monocultures (i.e., biodiversity effects on recovery) for drought-

selected plants were more positive than those for ambient-selected plants (Fig. 3b; Supplementary Table 5). This is consistent with the more positive biodiversity effects on productivity for drought-selected plants after the subsequent drought event described in the previous section. However, biodiversity effects on resistance were more negative for drought-selected than for ambient-selected plants, thus leading to similar biodiversity effects on resilience between the two selection treatments (Fig. 3a, c).

**Plant interactions.** We measured plant interaction intensity, $NInt_M$[44], by comparing plant biomass between individual-plant pots and multi-plant pots (Eq. (2) in "Methods"), separately for monocultures and mixtures. The interaction intensity was mostly negative (Supplementary Fig. 4), indicating that plants in pots

with four individuals (monocultures or mixtures) had less biomass than plants in pots with one individual. Nevertheless, in most cases average individuals in pots of four individuals had more than 25% of the biomass of individuals in pots with one individual, that is $NInt_M > -0.75$, the value expected under the reciprocal yield law[45]. An exception was the monocultures after the drought event, which produced the same amount of biomass per pot independent of the number of plants ($NInt_M \approx -0.75$; Supplementary Fig. 4). Instances of facilitation, i.e., cases where individual plants in pots with four individuals had more biomass than individual plants in pots with one individual and $NInt_M > 0$, were very rare.

During the drought phase in the glasshouse, drought-selected plants competed ($NInt_M < 0$) more strongly in mixtures than did ambient-selected plants (Supplementary Fig. 4; Supplementary Table 6), which was also the time when drought-selected plants tended to have lower complementarity effects than ambient-selected plants (Fig. 2e). After the subsequent drought event, drought-selected plants showed weaker heterospecific than conspecific competition, which was less pronounced for ambient-selected plants (Fig. 4; Supplementary Tables 7 and 8), consistent with the positive net biodiversity effects on productivity and recovery reported above for drought-selected plants after the subsequent drought event. These results suggest that plants whose ancestors were exposed to recurrent droughts had reduced interspecific relative to intraspecific competition (i.e., increased niche differentiation), at least when growing in mixtures and after a subsequent drought event.

**Plant traits.** We measured six traits that were closely related to plant water or carbon use on plants from pots with one individual. Drought- and ambient-selected plants had similar average values of leaf relative chlorophyll content, leaf area (LA), leaf mass per area (LMA), and leaf osmotic potential before the drought event in the glasshouse (Supplementary Fig. 5). Species varied in their responses of LMA to the selection treatment (Supplementary Table 9). The subsequent drought event reduced leaf stomatal conductance (Supplementary Fig. 6). However, leaf stomatal conductance did not vary significantly between the two drought-selection histories, neither before nor during the subsequent drought event (Supplementary Fig. 6). These results

**Table 1 Significance tests for the effects of selection treatment on net biodiversity effect (NE), complementarity effect (CE), and sampling effect (SE) on productivity before, during, and after the drought event in the glasshouse.**

|  | df | ddf | F | P |
|---|---|---|---|---|
| *Before drought* |  |  |  |  |
| NE ($n = 255$) | 1 | 11.6 | 0.035 | 0.855 + |
| CE ($n = 254$) | 1 | 12.5 | 0.029 | 0.867 + |
| SE ($n = 254$) | 1 | 15.7 | 0.524 | 0.480 + |
| *During drought* |  |  |  |  |
| NE ($n = 254$) | 1 | 12.9 | 1.671 | 0.219 − |
| CE ($n = 248$) | 1 | 17.8 | 2.853 | 0.109 − |
| SE ($n = 248$) | 1 | 18.7 | 0.070 | 0.794 + |
| *After drought* |  |  |  |  |
| NE ($n = 219$) | 1 | 9.1 | 14.490 | **0.004** + |
| CE ($n = 188$) | 1 | 14.0 | 22.110 | **0.001** + |
| SE ($n = 188$) | 1 | 11.4 | 9.988 | **0.009** − |

Results are from mixed-effects analyses of variance by fitting block and selection treatment as fixed-effects terms and species composition and its interaction with selection treatment as random-effects terms.
Note: *df*, numerator degrees of freedom; *ddf*, denominator degrees of freedom (these reflect residual degrees of freedom among the selection responses of 15–21 species pairs [=species compositions] for which biodiversity effects were calculated). *F* and *P* indicate *F* ratios and the *P* values of the significance tests, respectively. Data in bold indicate significant results (*P* < 0.05). + or − besides the *P* values represents the direction of difference between drought vs. ambient-selection treatments. Numbers within brackets indicate the numbers of mixtures.

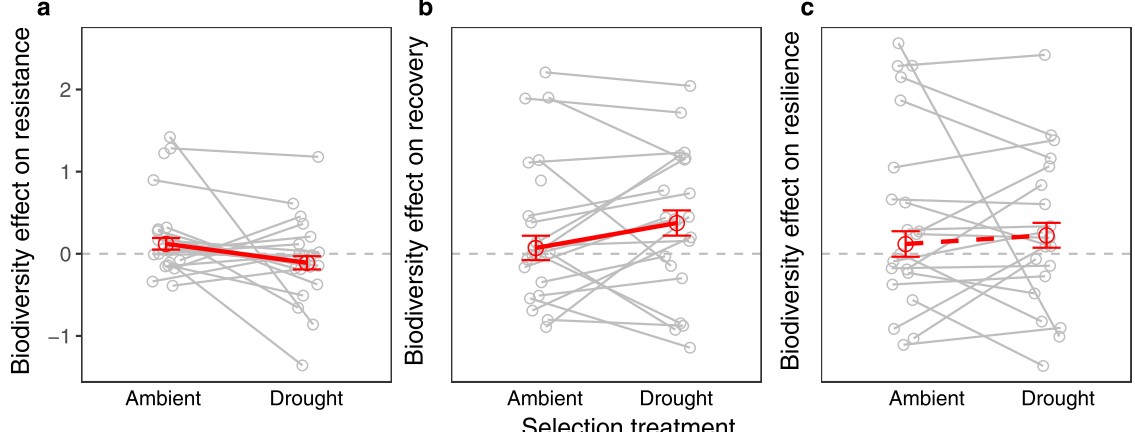

**Fig. 3 Difference in biodiversity effects on biomass stability in response to the drought event in the glasshouse between selection treatments (ambient and drought-selected plants).** The solid red lines indicate significant differences between the two selection treatments (see Supplementary Table 5). Biomass stability was calculated as resistance (**a**), recovery (**b**), and resilience (**c**). Biodiversity effects on stability were calculated as the differences in stability indices between mixtures and monocultures. Red points and error bars show the means ± standard error of 2-species mixtures of the corresponding selection treatment. Gray points represent means for species pairs (standard errors for species pairs not shown). Gray lines connect the same species pair between the two selection treatments. Source data are provided as a Source Data file. The numbers of mixtures per species pair (gray points) and per selection treatment (red points) are provided in the Source data file.

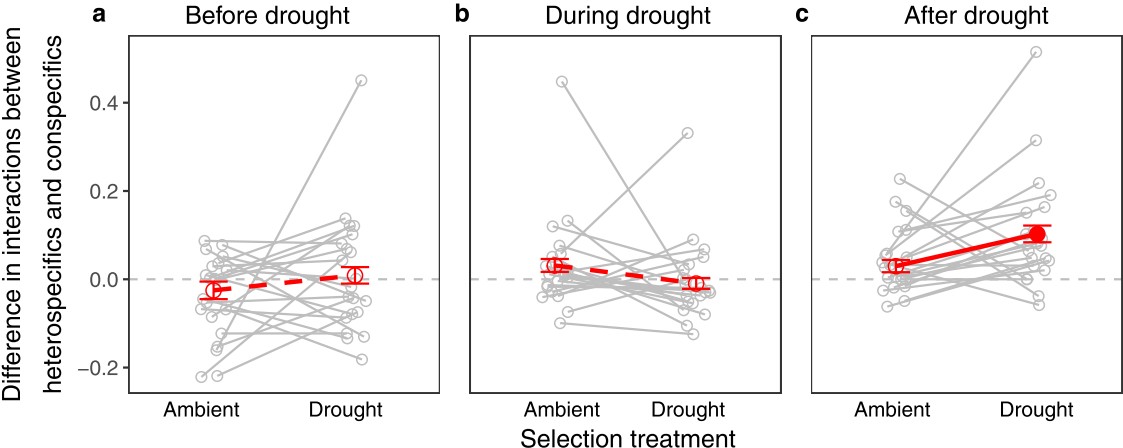

**Fig. 4 Effects of selection treatment (ambient- vs. drought-selected plants) on the difference between heterospecific and conspecific interactions.** The differences in plant interactions were calculated before (**a**), during (**b**), and after (**c**) the drought event in the glasshouse. The solid red line indicates a significant difference between the two selection treatments (see Supplementary Table 8). Red points and error bars show means ± standard error. The filled red point shows a significant difference from zero (horizontal dashed lines; see Supplementary Table 7), i.e., weaker heterospecific than conspecific competition. Gray points represent means for species pairs (standard errors for species pairs not shown). Gray lines connect the same species pair between the two selection treatments. Source data are provided as a Source data file. The numbers of mixtures per species pair (gray points) and per selection treatment (red points) are provided in the Source data file.

suggest that the selection treatment did not lead to significant changes in average values of leaf traits on plants grown alone, at least before the subsequent drought event. After this event, the time window for the biomass harvest after the drought did not allow us to take additional trait measurements. However, drought- and ambient-selected plants developed similar root-shoot biomass ratios after the drought event (Supplementary Fig. 7).

We also measured leaf relative chlorophyll content, LA and LMA on plants in mixtures before the drought event in the glasshouse. Drought-selected plants diverged more in LA between species in mixtures than did ambient-selected plants, although this difference was only marginally significant ($F_{1,19} = 3.66, P = 0.071$; Supplementary Fig. 8, Supplementary Table 10). However, species dissimilarities in LMA, leaf relative chlorophyll content, or the three traits jointly did not vary significantly between the two selection treatments (Supplementary Fig. 8, Supplementary Table 10).

## Discussion

Increasing threats from extreme climatic events such as droughts have raised the importance of predicting ecosystem responses to climate change, both in short and long terms, based on ecological and evolutionary theory[3-6,46-48]. In this study, we tested whether recurrent summer droughts caused transgenerational changes in species interactions and assessed the effects of drought-selection on biodiversity effects for 21 pairs of plant species over the time course of a subsequent experimental drought event (before, during, and after drought). We found that an 8-year treatment of recurrent droughts had caused transgenerational reinforcement of species complementarity in mixtures during the recovery phase after the subsequent drought event, which led to more positive biodiversity effects on community productivity and recovery after the drought event. These findings suggest that exposure to extreme climatic events during previous generations can improve the responses of offspring generations to future events in mixed-species grassland communities.

We found that drought-selection caused a significant difference (relative to ambient-selection) in biodiversity effects on productivity only after a subsequent drought event in the glasshouse. The more positive biodiversity effects in response to the drought

event for drought-selected plants were primarily due to more positive complementarity effects, whereas sampling effects were more negative than those of species pairs selected in the ambient treatment. The increased complementarity effects between drought-selected plants after the drought event was primarily the result of reduced interspecific relative to intraspecific competition (i.e., increased niche differentiation). In contrast to a previous grassland experiment with plants from the Jena Experiment selected under ambient conditions[37], we did not find evidence for facilitation underpinning complementarity effects. Facilitation, both within and between species, was very rare over the time course of the drought event in our case. During the most stressful phase of the drought event, drought-selected plants competed more strongly between species than did ambient-selected plants. This suggests that transgenerational effects on species interactions can differ between environmental conditions, and observations under ambient climates may not serve to make appropriate predictions for transgenerational responses of communities to extreme climatic events.

The increased niche differentiation for drought-selected plants after the drought event could be related to state shifts of different resources over the phases of the drought event. For example, droughts can increase leaf litter[49] and reduce the mobility of soil nutrients, the activity of soil microbes, and the rate of litter decomposition[50-52]. These constrained resources can be released after droughts[31,32,51], which may shift competition for a single resource (water) during the most stressful phase of drought to multiple resources after drought, thus increasing the potential for niche partitioning among species after drought.

In line with the finding of drought selection leading to stronger species complementarity during the recovery phase after the drought event, mixtures of drought-selected plants recovered faster than those of ambient-selected plants, which led to a more pronounced positive biodiversity effect on recovery for drought-selected plants. A previous field experiment[11] showed that more diverse communities were better able to compensate for drought-driven productivity loss, which led to stabilizing effects of biodiversity. The results from our study suggest that transgenerational effects of recovery from drought, the expression of which depends on levels of biodiversity, may be an important mechanism driving the compensatory recovery observed in the

field[11]. Biodiversity effects on resistance were more negative for drought-selected than for ambient-selected plants, thus leading to similar biodiversity effects on resilience between the two selection treatments. These findings suggest that drought-selected plants competed more strongly between species than did ambient-selected plants during the drought event, which might have impeded the resistance of mixtures relative to monocultures. Plants of the two selection treatments had similar resistance both in monocultures and mixtures. One explanation for this result could be that the recurrent summer droughts in the field did not cause severe mortality immediately during the droughts[53]. Thus, the importance of selection for traits associated with fast recovery from droughts may overwhelm that of selection for traits associated with high tolerance to droughts. The selection of acquisitive traits for fast recovery after droughts may have side effects on the processes during droughts, for example, intensifying species competition during droughts. Alternatively, the selection processes during droughts may have primarily played out belowground, which we could not formally test.

Positive biodiversity effects on productivity before a drought can lead to greater losses during a drought, thus reducing biodiversity effects and causing even negative biodiversity effects on resistance[13]. Negative biodiversity effects on productivity during a drought would require greater recovery after a drought, thus reducing biodiversity effects on recovery. These complexities imply that a full investigation over the time course of an extreme climatic event (before, during, and after an extreme climatic event), integrating both biodiversity effects on productivity and stability, is necessary for predicting the responses of ecosystems to extreme climatic events[9,11,54].

Both genetic and non-genetic processes may drive the observed transgenerational reinforcement of species complementarity. A previous study found that 11 years of selection by community diversity in the ambient treatment in the Jena Experiment caused the rapid emergence of populations with different genetic composition for three out of five perennial species[23], and one of the species (*Prunella vulgaris*) occurred in our study. Population genetic responses to the eight years of drought in the field might have been limited in some species due to small standing genetic variation at the beginning of the drought treatment in the field. In this case, mechanisms other than rapid evolution may dominate. Different mechanisms of transgenerational effects can also have mutual dependence. For example, DNA methylation was found to mediate adaptive, genotype-dependent effects of transgenerational plasticity in response to drought in an annual plant species[55]. Epigenetic mechanisms can also contribute to genetic evolution, if extreme climatic events become more frequent[19]. Planting more generations in common gardens and employing molecular analyses would empower future studies to better determine the heritability of, and the relative importance of, genetic vs. non-genetic processes to the transgenerational responses to extreme climatic events[19,20,23].

We could not determine which traits drove the increased species complementarity for drought-selected plants after the drought event. For the traits that we measured and analyzed, we did not find any significant difference between plants selected under recurrent droughts vs. plants selected in the ambient treatment. But we should note that this does not necessarily mean that the recurrent droughts did not select for specific traits or trait variation on plants growing in mixtures or after the subsequent drought event, because most of the traits were measured before this event and for single plants, while we detected the primary effects of drought-selection after the drought and in mixtures, a response that we had not anticipated. Measuring traits on single individuals may not capture the full selection consequences of drought, especially if drought-selection primarily induces changes

in species interactions, as we found here and has been suggested elsewhere[33,36,38], or if selection alters trait plasticity in response to interspecific neighbors. This is partially reflected by the findings that competing species composed of drought-selected plants diverged (relative to ambient-selected plants) in their leaf area before the subsequent drought event when they were growing together in mixtures. It is also possible that the drought-selection may be more apparent on root traits (e.g., root cortical thickness, hydraulic conductance, and mycorrhization rates) with close linkages to plant water acquisition, or reproductive traits (e.g., flowering time or seed production) associated with fitness[29,56,57], but again these were not measured in this study. A complete investigation of both above- and belowground traits in both monocultures and mixtures over the whole phases of a drought event (before, during, and after drought) would yield more insights on the effect of traits on transgenerational effects of increased species complementarity.

Although we only measured biodiversity effects and species interactions in mixtures with two species, these measures can provide fundamental insights into biodiversity effects in mixtures with more species, because productivity of mixtures with any level of species richness can be decomposed into the contribution from expected yields of corresponding monocultures and pairwise interactions between two species, if higher-order interactions are not important[58,59]. The overall effect of species interactions yields the net biodiversity effect. Furthermore, because there is a single interaction between species in species pairs, it is more straightforward to interpret this interaction than the multiple interactions occurring in more diverse mixtures.

In this study, we add an important mechanism, the transgenerational reinforcement of species complementarity, to explain the stability of productivity in mixed-species grassland communities facing climate change. The exposure to climate change during previous generations in grassland communities can increase species complementarity and thus improve the responses of offspring generations to future climatic events. Our results suggest that if past extreme climatic events do not completely exclude species, they may enhance the complementarity between species and the sustainability of biodiversity effects on ecosystem functioning in the face of future extreme climatic events.

## Methods

**Experimental design.** To test whether an 8-year treatment of recurrent summer droughts would change biodiversity effects and species interactions of grassland plants when facing a subsequent drought event, we grew ambient- vs. drought-selected plants of 12 species in a glasshouse. The plants were grown from seeds collected from 40 plots (Supplementary Data 2) under 8-year treatments of yearly summer droughts vs. ambient precipitation in a biodiversity field experiment in Jena, Germany[11,41].

The Jena Experiment was established in 2002 using a common seed pool of 60 grassland species, with 80 20× 20 m large plots of species richness levels of 1, 2, 4, 8, 16, and 60 species[40]. Most of the species are perennial and capable of outcrossing (Supplementary Table 1). The Jena Drought Experiment[11,41] was initiated in 2008. Two 1× 1 m subplots were set within each large plot, designated as either drought treatment or ambient control. For the drought treatment, rainout shelters were set up to exclude natural rainfall in mid-summer for 6 weeks. The ambient control treatment got the same shelter construction but rain water was reapplied to not confound the results with artifacts from the shelter[60]. We repeatedly harvested the aboveground biomass per year, once before and once after the summer drought treatment[11,41]. The design of the Jena (Drought) Experiment did not allow the exclusion of cross pollination or gene flow between subplots or large plots in the field. Such gene flow may have reduced the possibility for genetic differentiation and for the observed effect sizes of the selection treatment[23]. We collected seeds from drought and control subplots throughout the 2016 growing season (Fig. 1). We obtained seeds of 17 species, but only used 12 of them, because the other five species had either few seeds or low germination rates. Seeds per species per selection treatment were collected from 4 to 23 (interquartile range: [8.50, 17.00]) maternal plants distributed across 2–10 (interquartile range: [4.75, 9.00]) large plots in Jena Experiment, in which the functional group richness ranged from 1 to 4 (Supplementary Data 2). The 12 plant species represented four functional groups (grass, small herb, tall herb, and legume) (Supplementary Table 1). The

detailed classifications of the functional grouping can be found in the design of the Jena Experiment[40]. Eleven of the 12 species were perennial, and one was annual (*Trifolium dubium*). The average longevity of the perennial species in the Jena Experiment has been estimated at 3–4 years[61], so that multiple generations and sexual reproduction cycles could occur during the 8-year drought treatment. Although each subplot was small, population sizes of each species were estimated to range from 100 to 1000 individuals m$^{-2}$ in ambient and drought subplots at the beginning of the drought treatment in the field[62].

We germinated the seeds in Petri dishes and transplanted the seedlings into pots in February 2017 in a glasshouse (day temperature range 20–25 °C, night temperature range 15–21 °C, and humidity range 60–80%) at the University of Zurich, Switzerland. Seedlings were planted individually, in monocultures, or in 2-species mixtures in the glasshouse (Fig. 1). In the glasshouse experiment, both monocultures and mixtures contained four plants within a pot. The pots were 11×11×11.5 cm in size and filled with soil composed of 50% collected from a sugar-beet field, 25% sand and 25% perlite. We randomly assigned the pots into four blocks in the glasshouse. To test the effects of drought-induced selection on plant traits, we planted individual seedlings of the 12 species in a fifth block. Within the first 2 weeks, dead individuals were replaced, thereafter dead individuals were not replaced anymore. In total, we established 958 pots: 257 pots of mixtures, 217 pots of monocultures, and 484 pots of individual plants (244 pots of individuals in blocks 1–4, and 240 pots of individuals in block 5; Supplementary Methods). For mixtures, there were 21 species pairs (Supplementary Table 1). Species pairs composed of *Crepis biennis* or *Lotus corniculatus* had low numbers of replicates (Supplementary Table 1). However, including or excluding these communities produced qualitatively similar results. Thus, we present the results including these two species pairs in this paper. We provide detailed explanations on the choices of species pairs and regarding the biodiversity treatment history in the Jena Experiment in Supplementary Methods.

During a first phase of 3 months in the glasshouse (Fig. 1), pots were watered regularly ("before drought"). After 14–16 weeks, when most of the species had reached peak aboveground biomass, we harvested all individuals in each pot by cutting them 3 cm above the ground, allowing regrowth from the left plant bases (first harvest, "before drought"). The time span for the first harvest included both the time for trait measurements (section "Plant traits" below) and for the immediately following biomass harvest. We completed the biomass harvest of each block within 1–2 days. This allowed us to account for the larger time differences between blocks by fitting block effects in the statistical analyses. After the first harvest of each block, plants were watered regularly and allowed to regrow until the 18th week from planting. This was followed by a second phase of 2 weeks without watering. Soil moisture decreased from more than 40% to less than 10% after 10 days since drought initiation. At the end of the second phase, that is after 20 weeks from planting, we made a second aboveground harvest at 3 cm above the ground (second harvest, "during drought"). During a third phase of 7 weeks, pots were watered regularly again for recovery until most plants reached a new aboveground biomass peak again. At the end of the third phase, that is after 27 weeks from planting, we harvested both above- and belowground plant biomass (third harvest, "after drought"). We checked and confirmed that most plants had reached the full-grown state and peak biomass before each harvest by monitoring their flowering. After each harvest, we cleaned and dried the harvested plant material at 70 °C for 48 h to obtain the dry biomass. We used the aboveground biomass as a proxy for productivity. Although clipping may affect plant responses to the experimental drought in the glasshouse, clipping had the advantage that all plants were "standardized" in height before the experimental drought, thus reducing carry-over effects of differential growth before the experimental drought.

**Additive partitioning**. We used the additive partitioning approach (Eq. 1)[17] to decompose the net biodiversity effect (NE) on aboveground biomass into the complementarity effect (CE) and the sampling effect (SE):

$$\triangle Y = Y_O - Y_E = N \overline{\triangle RY} \bar{M} + N \, cov(\triangle \mathbf{RY}, \mathbf{M}), \qquad (1)$$

where $\triangle Y$ is the NE; $Y_O$ is the observed yield (productivity) in a mixture; $Y_E$ is the expected yield in the mixture, calculated from the observed yield in monocultures and their corresponding species proportions planted in the mixture, here 0.5; the two additive terms at the right side of the equation represent CE and SE, respectively; $N$ is the number of species in the mixture, here 2. The partitioning is based on the observed and expected relative yield (RY) of species in the mixture. The expected RY of species in the mixture is the proportion planted. $\Delta \mathbf{RY}$ is the difference between observed and expected RY of species in the mixture; $\overline{\Delta RY}$ is the average of $\Delta \mathbf{RY}$. A positive $\overline{\Delta RY}$ indicates a positive CE; a positive covariation between monoculture yield (**M**), and $\Delta \mathbf{RY}$ indicates a positive SE. More details about the calculation can be found in Loreau and Hector[17]. We conducted the partitioning separately for each harvest, selection treatment, and block. We did not perform the partitioning for mixtures with zero biomass[63]. For monocultures with zero biomass in the second or third harvest, we kept the ones which had positive biomass in the previous harvest but excluded the ones which had zero biomass in the previous harvest. For example, when performing the partitioning for the second harvest, we kept the monocultures that had zero biomass in the second harvest but non-zero biomass in the first harvest; we excluded the monocultures that had zero

biomass already in the first harvest. This was to assure that communities that died before the drought could not reappear during or after the drought, and communities that had died during the drought could not reappear after the drought.

We used mixed-effects models to assess the influences of drought vs. ambient-selection treatments on biodiversity effects (NEs, CEs, and SEs) separately for each harvest (Fig. 2; Table 1). Block and selection treatment were set as fixed-effects terms, while species composition (identity of species pair) and its interaction with selection treatment were set as random-effects terms. This conservative approach was used to allow for generalizations across all possible species compositions, although a more liberal approach with species composition and its interactions as fixed-effects terms could also have been applied (see Schmid et al.[64] for a discussion of defining terms as fixed- vs. random-effects terms, including a justification of preference for treating block as a fixed-effects term). We square-root transformed the CEs and SEs with sign reconstruction ($sign(y) \sqrt{y}$) prior to analysis to improve the normality of residuals[17]. The mixed-effects model did not converge in the analysis with CE after the drought event. In this case, we used a general linear model, in which we fitted block, species composition, selection treatment, and species composition by selection treatment interaction in this order. Then we tested the significance of selection treatment using its interaction with species composition as an error term. This procedure is an alternative to mixed-effects models that estimate variance components for random-effects terms with maximum likelihood[64].

To test whether biodiversity effects on productivity differed from zero, we additionally tested the significance of NEs, CEs, and SEs separately for each selection treatment and harvest (Supplementary Table 3). We set block and species composition as fixed- and random-effects terms, respectively. The model corresponding to CE for ambient-selected plants during the drought event did not converge so that we fitted it with a general linear model, in which we tested the significance of the overall mean (intercept) using species composition as an error term. All statistical analyses were conducted in R 3.6.3[65]. The mixed-effects models were conducted with asreml-R package 4.1.0.110[66].

Finally, we also tested whether the effects of drought selection on biodiversity effects (NEs, CEs, and SEs) in the glasshouse depended on the history of biodiversity treatment in the Jena Experiment. Most plants in the 2-species communities in the glasshouse originated from mixtures in the Jena Experiment (Supplementary Data 2; whether mixtures in the glasshouse composed of plants originating from monoculture field plots did not affect the effects of drought-selection on biodiversity effects on productivity (Supplementary Data 3)). To increase statistical power, we used functional group richness, ranging from 1 to 4, instead of species richness of the field plots as explanatory variable (Supplementary Methods). We fitted functional group richness either in linear (Supplementary Data 4) or log-linear (Supplementary Data 5) form. We did not detect significant effects of field treatment of functional group richness nor significant interactions between field treatment of functional group richness and the drought-selection history. Therefore, we excluded the history of biodiversity treatments in the field from further analyses.

**Biomass stability to the drought event in the glasshouse**. To assess the temporal responses of community aboveground biomass to the drought event, we calculated three indices representing different facets of stability: biomass resistance, recovery, and resilience (see van Moorsel et al.[43] for an example). We calculated resistance as the biomass ratio during vs. before the drought, recovery as the ratio after vs. during the drought and resilience as the ratio after vs. before the drought (see also Isbell et al.[9]). We log-transformed the indices (plus a half of the minimum positive value to allow taking logs of indices that were originally zero) prior to statistical analyses to improve the normality of residuals. Excluding index values that were originally zero produced qualitatively similar results.

To assess the effects of drought-selection on biomass stability, we fitted mixed-effects models with block and selection treatment as fixed-effects terms, and species composition and its interaction with selection treatment as random-effects terms (Supplementary Fig. 3; Supplementary Table 4). We fitted the models separately for mixtures and monocultures. We included the log-transformed biomass at the first harvest as a covariate because biomass stability in response to droughts often depends on plant performance under ambient conditions.

In the same way as net biodiversity effects on productivity were calculated for additive partitioning, we calculated biodiversity effects on biomass stability as the difference between each mixture and its corresponding monocultures. Then, we tested the influence of selection treatment on the biodiversity effects on biomass stability. Block and selection treatment were set as fixed-effects terms; species composition and its interaction with selection treatment were set as random-effects terms (Fig. 3; Supplementary Table 5). The log-transformed biomass at the first harvest was also included as a covariate[43]. To assess the significance of biodiversity effects on biomass stability for each selection treatment, we fitted another set of simplified models, with block and log-transformed biomass as fixed-effects terms, and species composition as random-effects term (Fig. 3).

**Neighbor interactions**. We assessed interactions between neighboring plants within pots using the metrics of neighbor interaction intensity with multiplicative

symmetry ($NInt_M$)[44]:

$$NInt_M = 2 \frac{\triangle P}{P_{-N} + P_{+N} + |\triangle P|},\qquad(2)$$

where $P_{-N}$ and $P_{+N}$ are the productivities without (individual plant) and with neighbors (monocultures or mixtures), respectively; $\triangle P = P_{+N} - P_{-N}$. Negative values of $NInt_M$ indicate competition and positive values indicate facilitation. $NInt_M$ is bounded between –1 (competitive exclusion) and 1 ("obligate" facilitation). For monocultures, we first calculated the per-plant biomass as the ratio between total biomass and planting density, and then used the per-plant value to compare with the corresponding individuals (without neighbor) of the same species with the same selection treatment in the same block. Note that under the reciprocal yield law[45], an individual grown alone in a pot should be four times larger than an individual grown with three others in a pot, resulting in a $NInt_M$ of –0.75. For 2-species mixtures, we calculated the per-plant biomass separately for each species and took the average $NInt_M$ of the two species to measure the interaction intensity of the mixture. We set zero biomass for dead plants in the calculation. Again, if mixtures would also follow the reciprocal yield law independent of species identity, then $NInt_M = -0.75$ would be expected. Values greater than –0.75 indicate some sort of overyielding due to higher density or higher density and higher diversity.

To assess how selection treatment modified interactions between plants, we tested the effects of selection treatment on neighbor interaction intensity separately for monocultures and mixtures. We included block and selection treatment as fixed-effects terms, species composition and its interaction with selection treatment as random-effects terms (Supplementary Fig. 4; Supplementary Table 6).

We calculated the difference between the heterospecific interaction in a mixture and the conspecific interactions in its two corresponding monocultures. A positive value of this difference indicates a weaker heterospecific than conspecific competition (i.e., niche differentiation) or stronger heterospecific than conspecific facilitation, which may lead to a positive complementarity effect. We tested the effects of selection treatment on interaction difference for each harvest by fitting block and selection treatment as fixed-effects terms, and species composition and its interaction with selection treatment as random-effects terms (Fig. 4; Supplementary Table 8). We also tested the significance of the interaction difference for each selection treatment by fitting block and species composition as fixed- and random-effects term, respectively (Fig. 4; Supplementary Table 7).

**Plant traits**. To assess whether drought selection would change plant traits, we measured six traits (Supplementary Table 9) closely related to plant usages of water or carbon on plants in pots with one individual from blocks 1–5. We focused on the traits on individual plants without neighbor to evaluate the influence of selection treatment on traits without the impacts of plasticity induced by plant interactions. We measured leaf relative chlorophyll content, leaf area (LA), leaf mass per area (LMA) and leaf osmometric pressure before the drought; leaf stomatal conductance both before and during the drought; and dry biomass ratio between root and shoot after the drought (in the third harvest). Leaf relative chlorophyll content was measured for three mature, fully expanded leaves per plant by using a SPAD-502 Plus chlorophyll meter from Konica Minolta. LA was obtained by scanning 3–4 mature, fully expanded leaves per plant with a LI-3100C Area Meter from LI-COR. LMA was calculated as the ratio between leaf dry mass (oven-dried at 70 °C for 48 h, using the same leaves that for LA) and LA. Leaf osmotic potential at full hydration was considered as an important trait associated with plant tolerance to drought[30]. We measured leaf osmotic potential with freeze-thaw leaf pieces cut from 1 to 2 mature, fully expanded leaves per plant by using a Wescor vapor pressure osmometer VAPRO (Model 5520) according to the method by Bartlett, et al.[30]. Plants were fully hydrated 1 day before the leaf sampling for osmotic potential measurement. Leaf stomatal conductance is a measure of exchange rate of carbon dioxide and water vapor through the stomata[67]. It was measured for 3–5 healthy mature leaves per plant by using a SC-1 Leaf Porometer from Decagon Devices. For grass species, 3 blades were placed adjacent to each other to have a large enough area for the measurement of stomatal conductance. For stomatal conductance during the drought event, we measured the individual plants from block 5 only due to limited time during the drought phase. We harvested aboveground and belowground plant biomass separately for alive individuals at the end of the experiment (after the complete recovery from the drought). The oven-dried (70 °C for 48 h) aboveground and belowground biomass were used to calculate the biomass ratio between root and shoot. We took the average value of each trait of each plant for statistical analyses. Each trait was measured for each block in turn.

We used linear mixed-effects models to assess the influence (generalized across species) of selection treatment on trait values (red lines in Supplementary Figs. 5–7). Block and selection treatment were set as fixed-effects terms; species and its interaction with selection treatment were set as random-effects terms. Alternatively, we set species, selection treatment and their interaction as fixed-effects terms to assess whether species responded differently to the selection treatment (Supplementary Table 9). To test whether effects of selection treatment on traits differed between the five trait groups (leaf relative chlorophyll content, leaf area, leaf mass per area, leaf osmometric pressure, and leaf stomatal conductance) measured before the drought event in the glasshouse, we conducted

two alternative analyses. First, we performed a principal component analysis with all traits and retained the first two principal axes (PC1 and PC2), which accounted for 39.06% and 22.3% of the total variation, respectively. Then we used PC1 and PC2 as response variables in mixed-effect models, separately. We fitted the models with the same fixed- and random-effects terms as those using each separate trait as the response variable. Effects of selection treatment on PC1 or PC2 were not significant. Second, we pooled the five traits as a single response variable in a mixed-effect model (corresponding to multivariate analysis of variance). Block, trait group (a factor with five levels), selection treatment, and the interaction between trait group and selection treatment were set as fixed-effects terms; species and its interactions with trait group and selection treatment and their three-way interaction were set as random-effects terms. We did not detect significant effects of selection treatment nor its interaction with trait group. Therefore, we did not present the results associated with these multivariate analyses in this paper. LMA, LA, leaf osmotic potential, leaf stomatal conductance, and root-shoot biomass ratio were log-transformed to improve normality of residuals.

We also measured leaf relative chlorophyll content, LA and LMA in mixtures before the drought event (Supplementary Table 10) to evaluate the influence of selection treatment on trait dissimilarity between interacting species within communities. We calculated the absolute trait distance between two species in each mixture both separately for each trait and jointly with the three traits. For multi-trait-based dissimilarity, we standardized each trait to mean zero and unit standard deviation and calculated the Euclidean trait distance in standardized three-dimensional trait space.

We used linear mixed-effects models to assess the effects of selection treatment on trait dissimilarity in mixtures (Supplementary Table 10). Block and selection treatment were set as fixed-effects terms; species composition and its interaction with selection treatment were set as random-effects terms. The model for LA dissimilarity did not converge so that we fit it with a general linear model, in which we tested the significance of selection treatment using its interaction with species composition as an error term. For the models with LA, LMA, and the joint three traits as dependent variables, we removed one pot (B1P674) because the LA value of *Alopecurus pratensis* in this pot was extremely small (about 1/3 of the second minimum value of the same species in mixtures). However, including or excluding this pot produced qualitatively similar results.

**Reporting summary**. Further information on research design is available in the Nature Research Reporting Summary linked to this article.

## Data availability
The data supporting the findings of this study are available at the Figshare digital repository (https://doi.org/10.6084/m9.figshare.14511108.v1)[68]. Source data are provided with this paper.

## Code availability
The code supporting the results are available at the Figshare digital repository (https://doi.org/10.6084/m9.figshare.14511060.v1)[69].

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

## Acknowledgements

Y.C. acknowledges support from National Natural Science Foundation of China (32071536 and 31700349) and the Fundamental Research Funds for the Central Universities of China (20720210080 and 20720210075). B.S. was supported by the Swiss National Science Foundation (grant number 31003A_166457) and the University of Zurich Research Priority Program on Global Change and Biodiversity. N.E. acknowledges support from the German Centre for Integrative Biodiversity Research (iDiv) Halle-Jena-Leipzig, funded by the German Research Foundation (FZT 118). The Jena Experiment is funded by the Deutsche Forschungsgemeinschaft (DFG, German Research Foundation, FOR 456, FOR 1451, and FOR 5000). We thank Sofia J. van Moorsel, Gerlinde Kratsch, Ulrike Gudd and the gardeners of the Jena Experiment for help with the selection experiment.

## Author contributions

M.S.-L. and A.W. conceived the Jena Drought Experiment, Y.C., B.S., A.V., and C.W. conceived the glasshouse experiment to test selection effects. Y.C., M.I.-G., C.W., and T.X. carried out the glasshouse experiment. Y.C., B.S., and A.V. developed the analytical procedure, and Y.C. did the statistical analyses. Y.C. and B.S. wrote the manuscript with the help from N.E., M.I.-G., M.S.-L, A.V., A.W., C.W., and T.X.

## Competing interests

The authors declare no competing interests.
