## [Peer Review File · Nature Communications]

REVIEWER COMMENTS

Reviewer #1 (Remarks to the Author):

The study investigates how a 8-year drought treatment (drought vs. non-drought) modify the effect of biodiversity (i.e. growing multiple species together as opposed to monospecific stands) on productivity in the next generation, when plants are grown from seeds collected in drought plots vs. in ambient plots. A special focus is on the response (resistance, recovery, and resilience) of these second-generation plants to a drought event, and if this affected by the previous 8-year drought treatment . The authors found a greater biodiversity effect on community productivity and also on recovery from drought when growing plants from seeds collected in 8-year drought plots compared to plants from control plots. They suggest that this is an evolutionary response that helps communities to adapt to extreme climatic events.

Major comment:

1. My first concern is with the term “evolution”. It sounds strange to me that evolution takes place (to a detectable degree) within 8 years in a perennial grassland. We do not even know if in the 8-year experiment, there were dieback of individuals and establishment of new individuals (of the same species), or the individuals that were present after the 8 years, and which produced the seeds for the pot experiment”, are (1) the same individuals as in the beginning of the 8-year experiment, or (2) a subset of the individuals, or (3) a completely different cohort (generation) of the same species. As I think the study species are perennial (but this info is not provided), I would guess, that not much generational shift took place within the 8 years (but it would be interesting to read about this). It is possible that a selection process took place, and some phenotypes survived while others died out from a previously existing pool of different phenotypes. But I would not call this “evolution”, as this may have taken place within a single generation. By the way, an alternative explanation to evolution could be transgenerational plasticity.
2. I also have a serious concern related to the time span of the glass-house experiment. We read that seedlings were planted in February, and they were harvested after 16-20 weeks. I guess the study species are perennial species, so starting from seedling stage, 16-20 weeks sounds very short for perennial species of temperate grasslands to reach full-grown state. 16-20 week plants (starting from seedling stage) sound more like “juveniles”, and not like fully developed individuals. I may be wrong, but please add some more detail.
3. Another point is the biomass harvest of these juvenile (full-grown?) individuals. I know that biomass harvest is a standard method to estimate productivity/aboveground biomass, but the harvest regime presented (three times a year) sounds very intense and it is hard to imagine that this does not strongly affect the further growth of these (juvenile) plants. We read that all plant parts are removed from above 3 cm. How does this affect the plants (and the species of different growth form are affected very differently), and especially the further growth potential? The first harvest is done just before the dry period, which may also affect how plants are responding (to drought and to harvest). Please provide some justification that this harvest regime does not affect plants too much, or at least that it does not affect treatment effects too much.
4. I think too many analyses and results are presented. I would suggest that the manuscript should focus on the most important results. To me, these are Fig 3 and Fig 4. I would consider putting Fig 2 into Appendix, and I would possibly leave out (or narrow down) the part on the effects of drought selection

on plant interactions, and perhaps even the plant trait part as well (not much change in plant trait values, and no Figure on this latter section is presented in the main text).

5. it is kind of strange that we read in the Abstract and in the Introduction that the effect of biodiversity on buffering against extreme climatic events is well-established, this present study does not find such a pattern. Although there is a biodiversity effect on productivity in the first period (before the drought), there is no biodiversity effect (the effect is not different from zero) during the drought period, during the post-drought period, and also in the case of resistance, recovery, and resilience (in some cases there are differences between ambient and drought treatment, none of them is different from zero).

Minor remarks

6. line 101. What do you mean by “novel” drought? Novel relative to what?

7. l.104. 12 species were planted in 2-species mixtures. In all combinations? $(12 \times 11)/2 = 66$ species pairs. All of them were created? If, yes, this information should be provided. If not, then how species combinations were created? We read (l. 393) that “257 pots of mixtures” were used, but how this number came up, and how species combinations were created?

7. l. 117 “complementary effect” and “sampling effect” are mentioned, but it is not clear what these terms exactly mean, and how they were calculated (this is also missing from the Methods part (l. 418, 419))

8. l. 238. “21 pairs of plant species” Where does this number come from. I did not see this number in the Methods section (perhaps I missed it), how does this come from the 64 potential combinations (from 12 species).

9- l. 337 “what is “quarter quantile”. Perhaps “interquartile range”?

10. l. 372 “seeds were collected from mixtures” Does this mean the 60-species mixtures?

11. l. 377 Are these species perennial ones?

12. l. 384. What is a “seed family”?

13. l. 402. First harvest took place 14-18 weeks after planting seedlings. Why is this range in time span, and why not a single standard length? I think time span can be very important regarding how big a plant can grow from seedling stage.

14. Fig 1. Experimental design. It is kind of misleading that the horizontal axis is generally time, but when it gets to mixture types (Mixture/Monoculture/Individual), it is not a temporal series (one might think it is). It now seems that “Mixture” is under „Ambient”, Monoculture” is under „Drought”, and „Individual” is under „Ambient”. I would consider putting the three mixture types vertically.

15. Fig 2. I would consider putting this into Supplementary Figures

16. I am also not sure, if Fig 5 and Fig 6 are needed, and in the main text =see my Major comment No. 4

Reviewer #2 (Remarks to the Author):

In this paper, authors analyze the interactive effects of drought adaptation and biodiversity on the response of plant species to drought. For doing this, they used seed sources of 12 different species subjected to eight recurrent summer drought vs control, grew them under controlled conditions in monocultures or species mixtures applying an additional dry period and analyzed the effect of previous drought and biodiversity in terms of biomass productivity.

The experimental approach was carefully conducted, and the analytical approach is sound. I really liked the paper, that is well written and discussed. The separate analysis of effects at three different times (pre-, during and post-drought) is really interesting and brings novel results, and the analysis of combined effects of drought adaptation and biodiversity is a timely and interesting research topic. However, I have found a couple of issues that prevent the extraction of wide conclusions from the experiment. First, the use of biomass production as a single indicator of community performance is limited to some extent. Although productivity is an important ecosystem feature, it is not always related to individual fitness. In my opinion, it would be much more interesting to continue the experiment some additional weeks in order to have data on flowering and/or seed production. This additional information would have complemented the presented results with a more reliable determination of species-specific effects of the studied variables.

Second, I am not completely sure about the consequences extracted from the “biodiversity effect”. Authors compare the response of species growing in monocultures or with a different species. Although this approach gives a direct estimation of direct positive or negative effects between species pairs, these patterns would be completely different in communities with a higher number of species. For this reason, I am not sure whether authors can attribute the found interspecific effects to a “biodiversity effect”.

Reviewer #3 (Remarks to the Author):

The authors are interested in a very important topic that I agree is one of the most important environmental questions of our time, namely, (how) are species evolving with environmental stress and will evolved traits (or interactions) provide some level of stability in the face of monumental global change. The authors have used an existing long-term ecological experiment (The Jena Experiment) to test this question, have undertaken rigorous approaches to analyzing the data and have speculated on possible mechanisms, even though mechanisms they tested do not explain the patterns. They found that some mixtures of previously droughted plants show synergistic effects on aboveground biomass but only when subjected to a second drought event. This result is interesting, does suggest an evolutionary explanation, i.e., that something about previous droughting alters the ability of some plants when growing together to grow more, but unfortunately the design of the Jena Experiment does not allow for a test of evolutionary processes or selection – full stop. I appreciate all of the considerable work and thought the authors have taken on this paper and know that what they have done is a significant amount of effort undertaken by a lot of talented people. To that end, I have tried to explain my rationale and suggest other ways the data could be interpreted to address ecological patterns that this design does allow and suggest other edits to consider.

- The design of the Jena Experiment is good to allow examination of species-level consequences in patterns of varying mixtures to simulate differences in biodiversity. There is replication at the species level and at the functional group level (legumes, grasses, short/tall herbs as well as inherent differences in symbiotic microbiomes associated with each species (even though not examined here). What the experiment does not have is knowledge or control of seed family or even seed origin which allows for intraspecific or population-level variation (where you get replication to address selection). With no information of where the original seed sources came from, no population-level differences in seed origin

or even different vendors there is no ability to say with any statistical rigor that there are differences in droughted vs. ambient plants, you have an $n=1$. In quantitative genetics studies, seed family or population are critical to showing that there is variation to allow one to test – and analyze – that selection has occurred. The fact that there are no differences in any of the traits that were measured suggest that there was simply not enough genetic variation to begin with for drought as a selective agent to act upon. additionally, my understanding of the design of the Jena Experiment is that there is also complete gene flow (i.e., there is no barrier between monocultures or mixtures or between droughted and ambient plots) which means that genes are mixing at all levels. Under these conditions you simply have no ability to determine if and how selection occurred and no real mechanism for why it would have occurred, excepting drift (which could be tested but not with this design, due to the same reasons above).

- That said, the pattern that was documented is interesting and worth exploring more in an ecological context. When reading this paper I wanted to know more about the specific species that showed complementarity effects when grown in mixture under a 2nd drought treatment. Were there any common species or functional groups or mycorrhizal status (even based on the literature) to suggest that some species just grew better together? Having species or functional group (or a new category based on mycorrhizal/symbiotic partners status) in the models as fixed effects, as well as some interesting co-inertia analyses, might allow you to determine something more specific about the complementarity effects due to which species are interacting. There is a large and growing lit on how mycorrhizal or microbiome interactions can lead to enhanced water uptake in plants and there may be some insight to be gained by focusing more on the biology of the specific interacting plants in this study that show synergism when growing together in a 2nd drought treatment.

- An alternative explanation for this pattern is some kind of drought induced epigenetic effect. There is a growing literature on how up-regulation of water use efficiency or uptake genes may lead to enhanced growth that is worth more discussion (even though you did not examine this).

- Regardless of how you choose to revise this paper, care should be taken to better define specific hypotheses and match analyses to these hypotheses. Currently, all objectives are organized around analysis of the biomass to test different stability responses without linking directly to specific hypotheses.

This work is a huge effort and I urge the authors to find other ecological ways to interpret these data, since the design does not allow for the evolutionary approaches the authors want to use.
All the best.

Reviewer #4 (Remarks to the Author):

In the manuscript “Evolution of species complementarity in response to drought in a grassland biodiversity experiment”, the authors describe a study in which they investigate how selection in response to recurrent drought vs. ambient rainfall (8 years) in a grassland biodiversity experiment

influences biodiversity effects on productivity in the stages before, during, and after a drought. The authors found that evolutionary responses to drought resulted in more positive biodiversity effects on productivity post-drought, driven by greater recovery from drought and greater complementarity between species. The authors also measured a suite of functional traits to determine the basis for these effects but found that generally these did not explain the observed results. I found the research questions to be novel, interesting, and important. However, I have concerns about several aspects of the methods, including the method of measuring productivity, the seeds used in the experiment, the trait measurements. I also found that many ideas in the manuscript were not communicated clearly. My first concern is with the way productivity was measured. As I understand the methods as currently described, each plant was harvested after 14-18 weeks of ambient watering (first harvest), then harvested again after exposure to a two-week drought (second harvest), then harvested again after seven weeks of ambient watering (third harvest). Thus each plant was repeatedly clipped and allowed to regrow, as opposed to having different plants that were harvested at each different time point. It may be that I am misunderstanding the methods, but if plants are repeatedly clipped then this experiment is testing not just productivity responses to drought, but productivity responses to the interaction between drought and clipping.

Another concern is with the seed source and treatment of collected seeds. Firstly, seeds were collected from the field in 2016 and planted in the greenhouse 2017, such that there was no generation in between grown in common conditions to control for differences in maternal environments. As a result, it is possible that differences between drought and ambient precipitation plants are the result of maternal effects and not genetic changes. Secondly, seeds were collected from 1 x 1 m drought or ambient precipitation subplots within larger 20 x 20 m plots. This is a very small-sized area to test for evolutionary differences between the treatments; there is large potential for any evolutionary differences to be masked by gene flow from outside the subplots, particularly if the focal species are outcrossing (information about plant mating systems was not provided). Additionally, there is no information about how many plants of each species were present in subplots (i.e., the size of each potential evolving population). This small size of subplots makes it increasingly likely that maternal effects are at least partly responsible for observed treatment effects. There is a study mentioned in the discussion that shows a genetic basis for “adaptive changes” after 8 years of selection, but it is not clear what the selective regime was, what was measured, and whether those were the same seeds as used in this study.

Another concern related to the seeds used is that it's not stated from which diversity plots the seed originated. In the methods (L455-459), it is stated that the authors tested whether the effects of drought selection on biodiversity effects in the greenhouse depended on the functional group richness from Jena field plots, implying that seeds came from different plots. If this is the case, then it is possible that drought effects are confounded with diversity effects, but it is hard to know for sure because no information is given about the plots from which seeds came from and source plot was not included in the analyses.

A weakness of this paper is that functional traits were only measured on: 1) individuals growing in single-plant pots, or 2) individuals in mixtures before the drought (for a subset of traits - chlorophyll content, LA, LMA) and these were analyzed separately. A complete investigation of the functional basis of biodiversity effects before, during, and after the drought would require trait measurements on individuals growing in monocultures and mixed-species pots because evolved differences plasticity in response to heterospecific vs. conspecific neighbors may underlie differences between drought and

ambient selected plants. The analyses presented only examines mean trait values and so only presents part of the story. I believe this caveat should be included in the discussion at the very least.

Finally, while some parts of the manuscript are well written, there were many points throughout where things were not fully explained, worded confusingly, or used incorrect terminology. These instances are described in further detail in the specific comments.

My other comments and suggestions are specific to the following parts of the manuscript:

L40-41: "We tested this hypothesis..." - there isn't a hypothesis stated explicitly before this.

L45-46: "... showed greater between-species complementarity" - the meaning of this is somewhat obscure because it's not clear what exactly was measured on plants in the greenhouse.

L48-50: I don't think the results of this study demonstrate that biodiversity causes the evolution of increased complementarity between species in response to drought. The two selection treatments compared were recurrent drought vs. ambient precipitation - while seeds of the focal species from each of these two drought treatments were collected from mixed-species communities, there were no seeds collected from drought vs. ambient precipitation in less/more diverse communities to allow the effect of biodiversity per se to be tested.

L60-63: Each of these ecological mechanisms could be explained more completely as they are a bit obscure as currently written. For example, the mean and variance of what? Could you please define complementarity among species - i.e. is this niche partitioning that allows them to capture resources in a complementary manner in time or space? By what mechanism does higher biodiversity lead to increased recovery?

L68: "... via differential demographic processes" is vague and I think needs to be expanded on.

L71 (suggestion): "... select for [trait values that confer] higher resistance..."

L75: Why does nutrient availability increase after drought?

L79: The wording of "induce the evolution of species interactions" sounds like it is referring to the creation of new species interactions.

L81-88: This could be generalized to: climate change (drought) may increase or decrease niche overlap between two species depending on their respective shifts in phenotypic distribution. The distinction between tolerant and intolerant genotypes and species seems arbitrary and overly complicated. For example, in Supplementary Fig. 1a, there is no reason why the phenotypic distribution of the "tolerant" species might shift right but that of the "intolerant" species stays the same, leading to reduced niche overlap. The scenarios given here in panel a and b aren't the only possible outcomes within this framework.

L88-90: I'm not sure I follow this sentence. Does this refer to plasticity in trait expression resulting from interactions with neighboring species? If this is the case, how are these changes heritable? Some clarification is necessary.

L90-91: "Selection occurring at the community level" implies that biological communities are the units of selection. Also, is this sentence referring to selection traits in a species that is imposed by the surrounding community? Are the "individual" species mentioned here the species that constitute the surrounding community?

L107-108: "We harvested the biomass of every pot three times..." - after reading this and the methods it's still not clear to me exactly what was done. Was each harvest done on one of the four plants in each pot? Or were all four plants in each pot harvested, then allowed to regrow, then harvested again etc.? It seems like the latter from the way it is written, so if this is not the case then this needs to be made much clearer.

L116: Can you briefly define “biodiversity effects” here? The entire results section reports various biodiversity effects, and so a brief explanation of what positive/negative biodiversity (complementarity, sampling) effects represent would be useful.

L125-128: How did you calculate the intensities of interactions? By comparing the size of individuals in single-plant pots to those in multi-plant pots?

L131-134: I think the objectives should be before the methods overview so that readers understand why you are doing the things you describe there.

L133-134: “... investigate the roles of species interactions in driving the evolutionary changes in biodiversity effects” implies that species interactions are the selective agent leading to evolutionary change, whereas the selective agent evaluated was drought vs. ambient precipitation. Perhaps “investigate the roles of species interactions in mediating the evolutionary responses to drought that underlie biodiversity effects on productivity and stability”?

L143-150: Did you correct for multiple comparisons? If not, perhaps state how this compares to what would be expected just by chance.

L162: Change “positively” to “positive”.

L167: Change “plant” to “plants”.

L169: To me, this result indicates that the evolutionary shifts in drought-recovery are apparent only in certain contexts (i.e., in mixed-species communities).

L184: How is this index calculated?

L184-194: Is this paragraph only comparing individual plants to individuals in monoculture pots?

L192: Change “indications” to “instances”.

L195-196: How did you measure the strength of competition? Is this based on $N_{int}M$?

L227: “... was only marginally significant” – was it significant or not? The way it is written suggests that it was significant ($p < 0.05$).

L240, 248, 251, 267, 271, 329 etc.: “... had selected for increased niche differentiation”, “... selection for more positive biodiversity effects”, “... selection for increased complementarity”, “selection for niche differentiation” etc. – this is bit of a bugbear for me, and here’s why... To say that there was selection for something (e.g., increased niche differentiation, more positive biodiversity effects) means that that was the mechanism by which it increased fitness of individuals with those traits and so was favored by natural selection. While increased niche differentiation, for example, might be an effect of evolutionary responses to drought, it is not necessarily the cause of the evolutionary response (i.e., the mechanism by which individuals had higher fitness). We don’t know the mechanisms by which some individuals in drought plots had higher fitness; perhaps it was greater niche differentiation or perhaps it was something else, but this was never tested. To illustrate with an example, selection might favor greater rooting depth under drought because it plants to access deeper water (i.e., selection for a specific property that enhances fitness under drought – being more drought tolerant), which could then result in plants that differ more from other species in their rooting depth (i.e., selection of an object – plants with lower niche overlap with heterospecifics). The mechanism by which greater rooting depth increased fitness is because it allowed plants to exploit deeper water, the increase in niche overlap/biodiversity effects are just side effects. It is important to distinguish between the function of an adaptation (selection for) from the effect it has (selection of).

L268-270: This explanation of how drought can result in an increase in nutrient availability should be in the introduction where the idea is first mentioned.

L271: It’s not immediately obvious how a large flux of resources increases the potential for niche

partitioning. Could you cite a study that shows this effect?

L272: Why specifically epigenetic variation? Epigenetic variation is only one source of non-genetic variation.

L273: Explaining which results?

L273-276: What do you mean by adaptive changes? What was measured? And what was the selection regime? Are those seeds the same as the seeds used in this experiment?

L286: Again, “induces the evolution of species interactions” sounds like the creation of new interactions.

L276: I think discussion of trait results should be in its own paragraph.

L276-292: What about drought-induced evolution of plasticity in traits in response to the presence of neighboring plants? Could it be that measuring traits on single individuals (i.e., mean trait values) don't capture the full evolutionary consequences of drought?

L299-300: “... biodiversity-dependent evolutionary adaptation...” – this study never tested how different levels biodiversity influenced evolutionary adaptation as the study compared plants from drought vs. ambient plots. Perhaps “evolutionary responses to drought, the expression of which depend on levels of biodiversity” or something like that?

L320-326: This paragraph seems out of place and is not connected to the findings of this study.

L323: What do you mean by “a big picture”?

L336-339: This should be in the results.

L372: Seeds were collected from which mixtures? 60 species mixtures?

L377: What defines a “small” and “tall” herb?

L379: “... planted seedlings...” contradicts the later sentence “Seeds... were sown”.

L385: What do you mean by “perfect matches in seed families”?

L403: Did you harvest a single plant from each pot, or did you harvest the aboveground biomass of all the plants in a pot?

L417: What is the additive partitioning approach? Please explain this!

L420-422: “For the monocultures...” – I found this whole sentence to be quite confusing.

L429: What is the response variable in these models? NEs, CEs, and SEs?

L447: Why did you also run these separate treatment models?

L455-459: This is confusing, as earlier in the methods on L381-382 it is stated that “Seeds from drought and ambient plots of the same large plot in Jena were sown...”. This suggests that seeds weren't all collected from the same large plot, which I would assume have the same functional richness (as this is a plot-level property).

L558-565: With so many traits, it seems like it would be appropriate to use MANOVA first.

L570-572: How did you standardize trait values before calculating Euclidean distance?

Figures 2 and 3: Figures 2 and 3 more or less show the same information in different ways. I wonder if Fig.2 should be in the supplement rather than the main article as it seems slightly redundant.

Figures 5 and 6: As for Figs. 2 and 3, I wonder if Figure 5 is better in the supplementary info as it displays much of the same information as in Fig. 6.

Supplementary Table 1: Typo in “2-speices”

Reviewer #1 (Remarks to the Author):

The study investigates how a 8-year drought treatment (drought vs. non-drought) modify the effect of biodiversity (i.e. growing multiple species together as opposed to monospecific stands) on productivity in the next generation, when plants are grown from seeds collected in drought plots vs. in ambient plots. A special focus is on the response (resistance, recovery, and resilience) of these second-generation plants to a drought event, and if this affected by the previous 8-year drought treatment. The authors found a greater biodiversity effect on community productivity and also on recovery from drought when growing plants from seeds collected in 8-year drought plots compared to plants from control plots. They suggest that this is an evolutionary response that helps communities to adapt to extreme climatic events.

Major comment:

1. My first concern is with the term “evolution”. It sounds strange to me that evolution takes place (to a detectable degree) within 8 years in a perennial grassland. We do not even know if in the 8-year experiment, there were dieback of individuals and establishment of new individuals (of the same species), or the individuals that were present after the 8 years, and which produced the seeds for the pot experiment”, are (1) the same individuals as in the beginning of the 8-year experiment, or (2) a subset of the individuals, or (3) a completely different cohort (generation) of the same species. As I think the study species are perennial (but this info is not provided), I would guess, that not much generational shift took place within the 8 years (but it would be interesting to read about this). It is possible that a selection process took place, and some phenotypes survived while others died out from a previously existing pool of different phenotypes. But I would not call this “evolution”, as this may have taken place within a single generation. By the way, an alternative explanation to evolution could be transgenerational plasticity.

We have revised the interpretation of the major results throughout the paper within a broader context and refer to the observed transgenerational effects as “drought

memory” (see new title). We indicate that multiple non-mutually exclusive processes may contribute to the effects: rapid evolution, epigenetic effects and non-heritable parental effects. Although we did not test the relative importance of these three processes, we argue that rapid evolution (here defined as change in gene frequencies in populations) is possible in this study due to the following reasons. (1) Evolution by selection from standing genetic variation but not by epigenetic change was indicated in previous studies from the Jena Experiment (see Zupping-Dingley et al. 2014 and van Moorsel et al. 2019, references in manuscript (MS)). (2) Evolution by recombination (and mutation) during sexual reproduction cycles was possible considering an average longevity of 3–4 years for the perennial species in the Jena Experiment (Roeder et al. 2017, now cited in MS). Eleven of the 12 species were perennial and one was annual (*Trifolium dubium*). We have clarified this point in the Methods section (lines 477–480; Supplementary Table 1). (3) Population sizes were large, as we now report, with 1000 seedling per m² at sowing in 2002 (Roscher et al. 2004, cited in MS) and estimated to range from 100–1000 individuals m⁻² in ambient and drought subplots at the beginning of the drought treatment in the field based on previous records in the Jena Experiment (Marquard et al. 2009, cited in MS). We now describe this in more detail in the Methods section (lines 481–483). For the experiment in the glasshouse, we collected seeds from multiple maternal plants (interquartile range: [8.50, 17.00]) distributed across multiple plots (interquartile range: [4.75, 9.00]) in the Jena Experiment. Although we did not directly determine the standing genetic variation for the original seed pools at the beginning of the Jena Experiment, in the previous study of van Moorsel et al. (2019, cited in MS), we found a large genetic variation in all five perennial species tested there. However, we now also mention the caveat in the MS that population genetic responses to the 8 years of drought in the field might have been limited in some species due to small standing genetic variation at the beginning of the drought treatment in the field (lines 387–392). We have supplemented the information of seed sources in Supplementary Data 2 and discuss the possibility of evolutionary and other processes potentially explaining

the observed “drought memory” of species pairs tested in the glasshouse (lines 68–84, 378–400, and 465–468).

2. I also have a serious concern related to the time span of the glass-house experiment. We read that seedlings were planted in February, and they were harvested after 16-20 weeks. I guess the study species are perennial species, so starting from seedling stage, 16-20 weeks sounds very short for perennial species of temperate grasslands to reach full-grown state. 16-20 week plants (starting from seedling stage) sound more like “juveniles”, and not like fully developed individuals. I may be wrong, but please add some more detail.

We apologize that we were unclear about the developmental stage of our plants when we started to harvest them. We now point out that the plants at harvest were fully-grown individuals. Five of the 12 species had already fully developed flowers when measuring the traits before the drought treatment in the glasshouse, and most of the others at least started to flower at the time of the first biomass harvest. The earliest detection of fully developed flowers occurred about one month before the first harvest (species *Plantago media*). For plants before the second and the third harvests, we only recorded the occurrences of fully developed flowers for pots of single plants. We recorded three and six of 12 species that had fully developed flowers before the second and the third harvests, respectively. We did not record the occurrences of non-fully developed flowers before the last two harvests. But we checked and confirmed that most plants had reached a full-grown state and peak biomass before each harvest. Although all but one species were perennial, they all showed rapid growth and normally flower within the first vegetation season under beneficial conditions. We have clarified this point in the Methods section (lines 524–525).

3. Another point is the biomass harvest of these juvenile (full-grown?) individuals. I know that biomass harvest is a standard method to estimate productivity/aboveground biomass, but the harvest regime presented (three times a year) sounds very intense and

it is hard to imagine that this does not strongly affect the further growth of these (juvenile) plants. We read that all plant parts are removed from above 3 cm. How does this affect the plants (and the species of different growth form are affected very differently), and especially the further growth potential? The first harvest is done just before the dry period, which may also affect how plants are responding (to drought and to harvest). Please provide some justification that this harvest regime does not affect plants too much, or at least that it does not affect treatment effects too much. We apologized for this confusion. After the first harvest of each block, plants were watered regularly and allowed to regrow for about 2–4 weeks before the drought treatment. We did not start the drought treatment before we had checked and confirmed that all plants had regrown well. Thus, this harvest regime did not affect the further growth potential of plants. The time span of regrowth (2–4 weeks) was caused by the difference in the first-harvest time between blocks (14–16 weeks since the experiment). We have added this justification (lines 128–129, 505–516). We also mention now that these plant species are normally mown multiple times per growing season in the typical hay meadows from which they originated (up to four times in a management-simulation experiment in Jena, see Weigelt et al. 2009, cited in MS; lines 508–510).

4. I think too many analyses and results are presented. I would suggest that the manuscript should focus on the most important results. To me, these are Fig 3 and Fig 4. I would consider putting Fig 2 into Appendix, and I would possibly leave out (or narrow down) the part on the effects of drought selection on plant interactions, and perhaps even the plant trait part as well (not much change in plant trait values, and no Figure on this latter section is presented in the main text).

To simplify the presentation, we followed this advice and put Fig. 2 and the results associated with traits into the supplementary information and removed Fig. 5. We keep the results on species interactions in the main text, because one of the major objectives of this study was to investigate whether a history of drought selection can

modify species interactions in response to a new drought event. We have clarified the importance of the analyses on plant interactions (lines 104–108).

5. it is kind of strange that we read in the Abstract and in the Introduction that the effect of biodiversity on buffering against extreme climatic events is well-established, this present study does not find such a pattern. Although there is a biodiversity effect on productivity in the first period (before the drought), there is no biodiversity effect (the effect is not different from zero) during the drought period, during the post-drought period, and also in the case of resistance, recovery, and resilience (in some cases there are differences between ambient and drought treatment, none of them is different from zero).

We found significantly positive biodiversity effects on productivity before and after the new drought event, but not during this event in the glasshouse (Fig. 2; Supplementary Table 3). These dynamic biodiversity–productivity relationships across time may drive the non-significant biodiversity effects on stability indices (resistance, recovery, and resilience) (Pfisterer & Schmid 2002, cited in MS). Positive biodiversity effects on productivity before a drought can lead to greater losses during a drought, thus reducing biodiversity effects and causing even negative biodiversity effects on resistance. Negative biodiversity effects on productivity during a drought would require greater recovery after a drought, thus reducing biodiversity effects on recovery. We have added one paragraph in the discussion to explain these results (lines 353–361). We have also revised the introduction (lines 55–58) to show that the buffering effects of biodiversity for extreme climatic events can be dynamic across different phases of climatic events, which corresponds to the multiple dimensions of stability (e.g., resistance and recovery).

Minor remarks

We found that there are two comments ordered “7” in the minor remarks from Reviewer #1. We re-ordered the comments below to facilitate discussion across

comments.

6. line 101. What do you mean by “novel” drought? Novel relative to what?

We manipulated the “novel” (new) drought event in the glasshouse at Zurich. This drought treatment is “novel” (new) relative to the 8-year recurrent summer droughts in the field at Jena. We have clarified this point in the introduction (lines 104–106).

7. 1.104. 12 species were planted in 2-species mixtures. In all combinations?

$(12 \times 11) / 2 = 66$ species pairs. All of them were created? If, yes, this information should be provided. If not, then how species combinations were created? We read (l. 393) that “257 pots of mixtures” were used, but how this number came up, and how species combinations were created?

We only had a subset of the 66 combinations of 2-species mixtures due to the availabilities of seeds or seedlings, and a design based on the combination of functional groups (FGs). We collected seeds of 17 species in total but subsequently discarded two species with few seeds. We germinated the other 15 species, which included 6 short herbs, 3 tall herbs, 3 legumes and 3 grasses. We designed the species pairs based on these four FGs of the 15 species, using the following criteria. (1) Species pairs within each FG: for the FGs with three species (tall herb, legume and grass), we used all pairwise combinations ($n=3$); for the FG with six species (short herb), we created six pairs, in which each species occurred twice across the pairs. (2) Species pairs between FGs: for the combinations without short herbs, we randomly created three species pairs; for the combinations with short herbs, we randomly created 6 species pairs, in which species from short herbs occurred once across the pairs, and species from other FGs occurred twice across the pairs. This design yielded 42 species pairs in total. But after the germination, we found that three out of the 15 species had very low germination rates. Thus, we only used the 12 species with enough seedlings for the biodiversity experiment in the glasshouse, which resulted in 25 possible species pairs. Of these, we had to discard four species pairs due to

mortality during seedling transplantation, which resulted in the final number of 21 species pairs.

Next, we considered the two types of treatment histories in the Jena Experiment: drought and biodiversity. We treated the biodiversity history at the level of FG instead of species, because trait differences between species were much larger between than within FGs (Roscher et al. 2004, cited in MS), and manipulating history combinations of drought x FG richness allowed a more balanced design with larger potential effects than having history combinations of drought x species richness (which would also have been difficult due to limitations of space and labor resources). To separate the effects of drought selection history from that of biodiversity selection history, plants originating from Jena subplots with the same FG composition were combined for each drought selection history. With the above criteria and after exclusions of failed establishment, we obtained 257 pots of mixtures.

We have supplemented the above information in Supplementary Methods.

8. 1. 117 “complementary effect” and “sampling effect” are mentioned, but it is not clear what these terms exactly mean, and how they were calculated (this is also missing from the Methods part (l. 418, 419)

We apologize that we did not introduce the additive partitioning method more carefully. We have added this now, including the corresponding calculations and interpretations of these effects (lines 136–145; Equation 1 in Methods).

9. 1. 238. “21 pairs of plant species” Where does this number come from. I did not see this number in the Methods section (perhaps I missed it), how does this come from the 64 potential combinations (from 12 species).

As mentioned in more detail in our response to the 7th comment of Reviewer #1, we now explain more clearly how the number of 21 pairs of species came about (Supplementary Methods).

10- 1. 337 “what is “quarter quantile”. Perhaps “interquartile range”?

Corrected.

11. 1. 372 “seeds were collected from mixtures” Does this mean the 60-species mixtures?

We have added a table to show the information of seed sources (Supplementary Data 2). Seeds of each species of each drought selection history were collected from multiple maternal plants (interquartile range: [8.50, 17.00]) distributed across multiple plots (interquartile range: [4.75, 9.00]) with different levels of biodiversity in the Jena Experiment.

12. 1. 377 Are these species perennial ones?

All species were perennial except for the annual species *Trifolium dubium*. Including or excluding this annual species produced qualitatively similar results. Thus, we present the results including all species in this paper. We have added this information in the Methods section (lines 477–480) and Supplementary Table 1.

13. 1. 384. What is a “seed family”?

We have deleted this phrase.

14. 1. 402. First harvest took place 14-18 weeks after planting seedlings. Why is this range in time span, and why not a single standard length? I think time span can be very important regarding how big a plant can grow from seedling stage.

We checked and confirmed that most plants had reached a full-grown state and peak biomass before each harvest by monitoring flowering (lines 524–525; please see the response to the 2nd comment of Reviewer #1). We mis-assigned an additional two-week recovery time before the drought treatment to the time of the first harvest in the earlier version. The first harvest actually took place after 14–16 weeks, lasting for two weeks, because we needed so much time to complete the trait measurements and for

the immediately following biomass harvest. We completed the first biomass harvest of each block within 1–2 days. This allowed us to account for the larger time differences between blocks by fitting block effects in the statistical analyses. After the first harvest of each block, plants were watered regularly and allowed to regrow until the 18th week from planting. We have clarified the above information in the Methods section (lines 505–516).

15. Fig 1. Experimental design. It is kind of misleading that the horizontal axis is generally time, but when it gets to mixture types (Mixture/Monoculture/Individual), it is not a temporal series (one might think it is). It now seems that "Mixture" is under „Ambient“, Monoculture" is under „Drought“, and „Individual" is under „Ambient“. I would consider putting the three mixture types vertically.

Revised.

16. Fig 2. I would consider putting this into Supplementary Figures

Done.

17. I am also not sure, if Fig 5 and Fig 6 are needed, and in the main text =see my Major comment No. 4

We have deleted Fig. 5 to simplify the presentation. We kept Fig. 6 in the main text though, because it shows that it is niche difference rather than facilitation as the driving mechanism for the complementarity effects (please also see the response to the 4th comment of Reviewer #1). We have clarified the importance of the analyses on plant interactions (lines 104–108).

Reviewer #2 (Remarks to the Author):

In this paper, authors analyze the interactive effects of drought adaptation and biodiversity on the response of plant species to drought. For doing this, they used seed sources of 12 different species subjected to eight recurrent summer drought vs control, grew them under controlled conditions in monocultures or species mixtures applying an additional dry period and analyzed the effect of previous drought and biodiversity in terms of biomass productivity.

The experimental approach was carefully conducted, and the analytical approach is sound. I really liked the paper, that is well written and discussed. The separate analysis of effects at three different times (pre-, during and post-drought) is really interesting and brings novel results, and the analysis of combined effects of drought adaptation and biodiversity is a timely and interesting research topic. However, I have found a couple of issues that prevent the extraction of wide conclusions from the experiment.

Thank you for the helpful comments. We have revised the paper according to the suggestions and comments below.

First, the use of biomass production as a single indicator of community performance is limited to some extent. Although productivity is an important ecosystem feature, it is not always related to individual fitness. In my opinion, it would be much more interesting to continue the experiment some additional weeks in order to have data on flowering and/or seed production. This additional information would have complemented the presented results with a more reliable determination of species-specific effects of the studied variables.

We agree with the reviewer that having data on plant reproductive traits, such as flowering or seed production, would complement the presented results. Unfortunately, we only collected flowering state for a small subset of plants to monitor the maturity state of plant species. We did not collect seed production data. We had to find a

balance between assessing peak biomass production as an important fitness trait for perennial plants (only one of our species was an annual, where seed output clearly would be a better fitness measure) vs. assessing seed production (after peak biomass). We have added this limitation in the discussion (lines 416–424).

Second, I am not completely sure about the consequences extracted from the “biodiversity effect”. Authors compare the response of species growing in monocultures or with a different species. Although this approach gives a direct estimation of direct positive or negative effects between species pairs, these patterns would be completely different in communities with a higher number of species. For this reason, I am not sure whether authors can attribute the found interspecific effects to a “biodiversity effect”.

We agree that, ideally, we should have tested not only 2-species mixtures. However, considering the already complex design of the experiment, we focused on the 2-species level, because this is the simplest case of diversity treatment and easier to interpret than higher-order mixtures. Although we only measured biodiversity effects and species interactions in mixtures with two species, these measures can provide fundamental insights into biodiversity effects in mixtures with more species, because productivity of mixtures with any level of species richness can be decomposed into the contribution from expected yield of corresponding monocultures and pairwise interactions between two species, if higher-order interactions are not important (Kirwan et al. 2009 and Chen et al. 2020, cited in MS). The overall effect of species interaction yields the “net biodiversity effect”. We added one paragraph in the discussion to show the linkage between biodiversity effects measured in mixtures with two and more species (lines 425–434).

Reviewer #3 (Remarks to the Author):

The authors are interested in a very important topic that I agree is one of the most important environmental questions of our time, namely, (how) are species evolving with environmental stress and will evolved traits (or interactions) provide some level of stability in the face of monumental global change. The authors have used an existing long-term ecological experiment (The Jena Experiment) to test this question, have undertaken rigorous approaches to analyzing the data and have speculated on possible mechanisms, even though mechanisms they tested do not explain the patterns. They found that some mixtures of previously droughted plants show synergistic effects on aboveground biomass but only when subjected to a second drought event. This result is interesting, does suggest an evolutionary explanation, i.e., that something about previous droughting alters the ability of some plants when growing together to grow more, but unfortunately the design of the Jena Experiment does not allow for a test of evolutionary processes or selection – full stop. I appreciate all of the considerable work and thought the authors have taken on this paper and know that what they have done is a significant amount of effort undertaken by a lot of talented people. To that end, I have tried to explain my rationale and suggest other ways the data could be interpreted to address ecological patterns that this design does allow and suggest other edits to consider.

Thank you for the constructive comments and suggestions. We now try to better explain why we consider the design of the Jena Experiment suitable for a crude assessment of potential evolutionary responses to selection treatments—including references to previous studies from the Jena Experiment (Zupping-Dingley et al. 2014, van Moorsel et al. 2019)—even if not at the mechanistic level referred to by the reviewer. We acknowledge that non-genetic processes may also contribute to the findings in this paper, so that we have revised the title and interpretation of our results within a broader context, which involves multiple alternative processes leading to the observed transgenerational responses to drought. We have also supplemented new

analyses to focus our results in a more ecological context. Please see the detailed responses below. To facilitate the discussion across comments, we ordered the following comments of Reviewer #3.

1- The design of the Jena Experiment is good to allow examination of species-level consequences in patterns of varying mixtures to simulate differences in biodiversity. There is replication at the species level and at the functional group level (legumes, grasses, short/tall herbs as well as inherent differences in symbiotic microbiomes associated with each species (even though not examined here). What the experiment does not have is knowledge or control of seed family or even seed origin which allows for intraspecific or population-level variation (where you get replication to address selection). With no information of where the original seed sources came from, no population-level differences in seed origin or even different vendors there is no ability to say with any statistical rigor that there are differences in droughted vs. ambient plants, you have an $n=1$. In quantitative genetics studies, seed family or population are critical to showing that there is variation to allow one to test – and analyze – that selection has occurred. The fact that there are no differences in any of the traits that were measured suggest that there was simply not enough genetic variation to begin with for drought as a selective agent to act upon. additionally, my understanding of the design of the Jena Experiment is that there is also complete gene flow (i.e., there is no barrier between monocultures or mixtures or between droughted and ambient plots) which means that genes are mixing at all levels. Under these conditions you simply have no ability to determine if and how selection occurred and no real mechanism for why it would have occurred, excepting drift (which could be tested but not with this design, due to the same reasons above).

Important points. As mentioned in the response to the 1st comment of Reviewer 1, we could assume that standing genetic variation was present in the seed populations used to set up the Jena Experiment in 2002, and we are now citing van Moorsel et al. (2019) as evidence (lines 117–119, 381–387). We agree that our main replication is at

the level of species and species pairs. But even within each species, the observed effects are transgenerational and with reference to van Moorsel et al. (2019) more likely genetic than epigenetic. We agree that if evolution was involved, it would probably be mostly due to selection out of standing genetic variation (which nevertheless we consider “valid” evolution in the sense of changing gene frequencies in populations) and less so due to recombination or mutation (as now mentioned in the Methods section (lines 478–481), eight years corresponded to few sexual generations in these mostly perennial species). We now also mention that population sizes of each species could be estimated at around 100–1000 plants m⁻² based on other studies from the Jena Experiment (Marquard et al. 2009, cited in MS; lines 481–483), but we could not measure this directly, and we cannot totally exclude the possibility that random genetic drift acted in the same way in multiple plots and multiple species, thus producing significant results by chance (lines 465–468).

In detail, seeds of each species of each selection treatment history were collected from multiple maternal plants (interquartile range: [8.50, 17.00]) distributed across multiple plots (interquartile range: [4.75, 9.00]) in the Jena Experiment. There could be already large variation in genetic composition for original seed pools at the beginning of the Jena Experiment. A selection process could take place by filtering from standing genetic composition by differential survival and proliferation of genotypes (Barrett & Schluter 2008, cited in MS). For example, in the previous study of van Moorsel et al. (2019, cited in MS), we found a large genetic divergence between plants germinated from the original seeds (in 2002) of one tested perennial species in Jena. Eleven years of selection by community diversity had caused the rapid emergence of populations with different genetic composition for three out of five tested perennial plants in the Jena Experiment (van Moorsel et al. 2019, cited in MS), and one of the species (*Prunella vulgaris*) occurred in the current study as well. Second, evolution by recombination (and mutation) during sexual reproduction cycles was possible considering an average longevity of 3–4 years for the perennial species used in our experiment (Roeder et al. 2017, now cited in MS). All species were

perennial except for the annual species *Trifolium dubium*. Third, although the experimental design in Jena did not allow exclusion of gene flow between subplots or large plots, this would result in more similar populations, and would reduce the possibility to find any genetic difference and its consequence for biodiversity effects. But in this study, we detected significant differences in species interactions and biodiversity effects between drought- and ambient-selected plants after a new drought event. Fourth, although we did not detect any significant differences in traits between the selection treatments, this does not necessarily mean that the recurrent droughts did not select for specific traits or trait variation on plants growing in mixtures or after the drought event, because most of the traits were measured before the drought event in the glasshouse and for single plants, while we detected the primary effects of drought selection after the new drought event and in mixtures.

We acknowledge that our results could arise from multiple alternative processes such as evolution, epigenetic effects and non-heritable parental effects. We have revised the title and interpretation on the major findings within this broader background—transgenerational effects in responses to drought—throughout the paper.

We now discuss the possibilities of evolutionary processes and their linkages with other alternative processes in the Introduction (lines 68–84), Methods (lines 465–468, 478–483) and Discussion (lines 378–400), provide the information of seed sources in Supplementary Data 2, and discuss the limitations associated with the trait measurement (lines 401–424).

2- That said, the pattern that was documented is interesting and worth exploring more in an ecological context. When reading this paper I wanted to know more about the specific species that showed complementarity effects when grown in mixture under a 2nd drought treatment. Were there any common species or functional groups or mycorrhizal status (even based on the literature) to suggest that some species just grew better together? Having species or functional group (or a new category based on mycorrhizal/symbiotic partners status) in the models as fixed effects, as well as some

interesting co-inertia analyses, might allow you to determine something more specific about the complementarity effects due to which species are interacting. There is a large and growing lit on how mycorrhizal or microbiome interactions can lead to enhanced water uptake in plants and there may be some insight to be gained by focusing more on the biology of the specific interacting plants in this study that show synergism when growing together in a 2nd drought treatment.

Thanks for the inspiring comments. We conducted multiple new analyses to explore more about complementarity effects (CEs) in an ecological context. (1) To explore the importance of specific species pairs for CEs, we ranked the CEs of species pairs, separately for each harvest and selection treatment history. That is, we calculated the average CE of each species pair, and ranked them from the largest to the smallest (Supplementary Fig. 2). We found significant positive correlations between the CE rank of species pairs (i.e., consistent ranks of species pairs) before and during the new drought event in the glasshouse, for both ambient- and drought-selected plants (Supplementary Table 2). Species pairs of drought-selected plants reversed in their ranks in CEs (i.e., a negative correlation) before vs. after the new drought event (Supplementary Fig. 2). However, this reversal was not present for species pairs of ambient-selected plants (Supplementary Fig. 2). These results suggest that the history of recurrent droughts can cause a temporal trade-off between species complementarities under normal conditions vs. when recovering from a new drought. Using this analysis, we found no particular species pair that responded consistently more or less strongly to drought treatment history than did other species pairs with regard to CEs. (2) To explore the contribution of particular species to CEs, we tested whether the presence or absence of a specific species in species pairs would affect CEs, separately for each harvest and selection treatment history. We fitted block, species identity (i.e., presence) and species composition in order, and tested the significance of species identity using species composition as an error term. For drought-selected plants, the presence or absence of specific species in species pairs did not significantly change CEs after the new drought event (Supplementary Data 1).

These results suggest that the increased CEs after the new drought event of drought-selected plants were not due to particular species with large effects on CEs. (3) To explore the importance of mycorrhizal status for CEs, we obtained the mycorrhizal status of each species from the FungalRoot database (Soudzilovskaia et al. 2020. *New Phytologist* 227:955–966.). We found that all the 12 species are commonly associated with arbuscular mycorrhizal fungi and therefore could not test corresponding contrasts between pairs with different combinations mycorrhizal types of plants. But we agree that measuring root traits like mycorrhization rates may provide fundamental insights into the ecological mechanisms driving the observed complementarity effects. Now we mention this point in the Discussion section (lines 349–352, 416–420). Given the absence of clear differences between species pairs and between individual species, we did not explore further contrasts such as between legumes or grasses. Although it would of course have been nice to find such differences, the absence of these increases the generality of our findings, which were significant even though they were tested against the variation among species pairs as random-effects term. We now point this out more explicitly in the main text (lines 186–189).

3- An alternative explanation for this pattern is some kind of drought induced epigenetic effect. There is a growing literature on how up-regulation of water use efficiency or uptake genes may lead to enhanced growth that is worth more discussion (even though you did not examine this).

We now provide multiple alternative explanations for this pattern, such as rapid evolution, epigenetic effects and non-heritable parental effects (lines 68–84, 378–400). We now also discuss one case study of transgenerational effects caused by drought-induced DNA methylation (lines 393–395). We also refer to the above-mentioned previous study from the Jena Experiment, where we found that heritable changes after 11 years of selection in high vs. low diversity were due to genetic differences, with methylation differences following the genetic differences (van Moorsel et al. 2019, cited in MS; lines 381–387).

4- Regardless of how you choose to revise this paper, care should be taken to better define specific hypotheses and match analyses to these hypotheses. Currently, all objectives are organized around analysis of the biomass to test different stability responses without linking directly to specific hypotheses.

We have revised the paper to target our analyses at transgenerational effects (lines 68–84), which may arise from genetic or non-genetic processes. We have also defined our hypothesis, inspired by the stress gradient hypothesis in ecology, that plants from drought selection history should show less negative interactions and more complementarity between species than those from ambient selection history, leading to more positive biodiversity effects on ecosystem functioning (lines 109–114).

5- This work is a huge effort and I urge the authors to find other ecological ways to interpret these data, since the design does not allow for the evolutionary approaches the authors want to use.

All the best.

We have revised the title and interpretation of our results within a broader background, which involves multiple alternative processes leading to the observed transgenerational reinforcement of species complementarity in response to drought. Although we cannot distinguish the contributions from genetic vs. non-genetic processes, we justified the possibilities of evolutionary processes in this study (please see the response to the 1st comment of Reviewer #3). We have also supplemented new analyses to focus more in an ecological context—which specific species or species pair may make particularly important contributions to the increased complementarity effect for drought-selected plants after the new drought event in the glasshouse. We now present these analyses in the Results section (lines 179–189); however, we were unable to find any significant effects of individual species pair or species, leaving only our main findings that drought history increased complementarity effects across the range of tested species and species pairs as general results, which we find by itself

ecologically remarkable (please see the response to the 2nd comment of Reviewer 3).

Reviewer #4 (Remarks to the Author):

In the manuscript “Evolution of species complementarity in response to drought in a grassland biodiversity experiment”, the authors describe a study in which they investigate how selection in response to recurrent drought vs. ambient rainfall (8 years) in a grassland biodiversity experiment influences biodiversity effects on productivity in the stages before, during, and after a drought. The authors found that evolutionary responses to drought resulted in more positive biodiversity effects on productivity post-drought, driven by greater recovery from drought and greater complementarity between species. The authors also measured a suite of functional traits to determine the basis for these effects but found that generally these did not explain the observed results. I found the research questions to be novel, interesting, and important. However, I have concerns about several aspects of the methods, including the method of measuring productivity, the seeds used in the experiment, the trait measurements. I also found that many ideas in the manuscript were not communicated clearly.

Thank you for the detailed and helpful suggestions and comments. We have clarified the method of measuring productivity, the seeds used in the experiment, the trait measurements and the other methodological details. Please see the detailed responses below.

1. My first concern is with the way productivity was measured. As I understand the methods as currently described, each plant was harvested after 14-18 weeks of ambient watering (first harvest), then harvested again after exposure to a two-week drought (second harvest), then harvested again after seven weeks of ambient watering (third harvest). Thus each plant was repeatedly clipped and allowed to regrow, as opposed to having different plants that were harvested at each different time point. It may be that I am misunderstanding the methods, but if plants are repeatedly clipped then this experiment is testing not just productivity responses to drought, but

productivity responses to the interaction between drought and clipping.

First, we repeatedly harvested the plants to simulate the usual management of grassland in the native region. Traditionally, grasslands in this region are managed as hay meadows, and mown 2–3 times per growing season (May to September), even up to four times per growing season in one experiment within the Jena Experiment (Weigelt et al. 2009, cited in MS). Second, the repeated clipping also corresponds to the scenarios of the drought treatment and biomass harvests in the Jena Drought Experiment (Vogel et al. 2012 and Wagg et al. 2017, cited in MS), where we collected the seeds for the current experiment in the glasshouse. The biomass was harvested twice, once before and once after the summer drought in the Jena Drought Experiment (Vogel et al. 2012 and Wagg et al. 2017, cited in MS). Third, we left enough regrowth time for plants (most are perennial) before the start of the drought treatment in the glasshouse. The harvest did not affect the further growth potential of plants (please also see the response to the 3rd comment of Reviewer #1). We have clarified our choice of this harvest design in the Methods section (lines 463–465, 508–510). In addition, we mention the point of the reviewer that in this experiment plants were clipped as in normal grassland management in the region and that it is conceivable that unclipped plants would have responded differently to the experimental drought in the glasshouse (lines 528–529; because no plants were unclipped, we could not test for a potential interaction between clipping and drought treatment in the glasshouse). Clipping also had the advantage that all plants were “standardized” in height before the experimental drought in the glasshouse, thus reducing carry-over effects of differential growth before the experimental drought (lines 529–531).

2. Another concern is with the seed source and treatment of collected seeds. Firstly, seeds were collected from the field in 2016 and planted in the greenhouse 2017, such that there was no generation in between grown in common conditions to control for differences in maternal environments. As a result, it is possible that differences between drought and ambient precipitation plants are the result of maternal effects

and not genetic changes. Secondly, seeds were collected from 1 x 1 m drought or ambient precipitation subplots within larger 20 x 20 m plots. This is a very small-sized area to test for evolutionary differences between the treatments; there is large potential for any evolutionary differences to be masked by gene flow from outside the subplots, particularly if the focal species are outcrossing (information about plant mating systems was not provided). Additionally, there is no information about how many plants of each species were present in subplots (i.e., the size of each potential evolving population). This small size of subplots makes it increasingly likely that maternal effects are at least partly responsible for observed treatment effects. There is a study mentioned in the discussion that shows a genetic basis for “adaptive changes” after 8 years of selection, but it is not clear what the selective regime was, what was measured, and whether those were the same seeds as used in this study.

First, we agree with the reviewer that our results could arise from multiple alternative processes, including evolution, epigenetic effects and non-heritable parental or maternal effects. We have revised the interpretation of the major findings within this broader background—transgenerational effects in response to drought—throughout the paper. We also added more information about the species, which were perennial with one exception (lines 477–480; Supplementary Table 1). The breeding system of the species has not been investigated within the Jena Experiment, but most of them, to our knowledge, are outcrossing (line 457).

Second, although each subplot is small, the initial seedling density in 2002 in the Jena Experiment was 1000 m⁻² (Roscher et al. 2004, cited in MS), and in 2006 the number of plants was generally greater than 1000 m⁻² (Marquard et al. 2009), which we now report in the Methods section (lines 481–483). Based on the preceding numbers, we infer population sizes range from 100–1000 individuals per 1 m² per species in ambient and drought subplots at the beginning of the Jena Drought Experiment in 2008. We also mention now that in the study of van Moorsel et al. (2019) large genetic variation was found for five tested perennial species from the Jena Experiment. In their case, populations with genetically differential responses to

selection by the biodiversity treatment as assessed by DNA analysis were maintained in 1 m² plots in the experimental garden, which we also mention now (lines 117–119, 381–387). We also provide information about the number of maternal plants and of plots from which the seeds of each species were collected (Supplementary Data 2). Seeds of each species of each selection treatment history were collected from multiple maternal plants (interquartile range: [8.50, 17.00]) distributed across multiple plots (interquartile range: [4.75, 9.00]) in the Jena Experiment.

It is true that gene flow between ambient and drought subplots in the Jena Drought Experiment could have diluted effects of differential selection (lines 465–468). However, as we discussed with reference to previous selection experiments with plants from the Jena Experiment (Zurppinger-Dingley et al. 2014, van Moorsel et al. 2019), genetic differentiation most likely would have resulted from differential mortality and proliferation, i.e., selection on standing genetic variation introduced at initial sowing of the Jena Experiment (please see the responses to the 1st comment of Reviewer #1 and the 1st comment of Reviewer #3), whereas recombination during the estimated three sexual generations of the mostly perennial species (only one annual) probably contributed less. Finally, even if the observed effects would have mainly been due to other than genetic transgenerational effects, a possibility we now acknowledge, the consistency of these effects, namely the increased complementarity effects under the experimental drought in the glasshouse for plants taken from plots with eight years of drought history, in our view would be even more remarkable if due to seed carry-over effects working similarly across 21 species pairs.

In summary, we have supplemented the information of seed sources in Supplementary Data 2 and now discuss the possibilities of evolutionary processes and their linkages with other alternative processes in the Introduction (lines 68–84), Methods (lines 465–468, 478–483) and Discussion (lines 378–400).

3. Another concern related to the seeds used is that it's not stated from which diversity plots the seed originated. In the methods (L455-459), it is stated that the authors tested

whether the effects of drought selection on biodiversity effects in the greenhouse depended on the functional group richness from Jena field plots, implying that seeds came from different plots. If this is the case, then it is possible that drought effects are confounded with diversity effects, but it is hard to know for sure because no information is given about the plots from which seeds came from and source plot was not included in the analyses.

Thank you for this important point. We had actually tested this possibility but did not mention the results, because they were not significant. Seeds of the 12 species were from 40 large plots with different levels of biodiversity in the Jena Experiment (the seed source information now provided in Supplementary Data 2). Most plants in the 2-species communities in the glasshouse originated from species mixtures in the Jena Experiment. To separate the effects of drought-selection history from that of biodiversity-selection history, we manipulated each species pair in the glasshouse with plants originating from the Jena subplots with the same functional group richness, which ranges from 1–4 (Supplementary Methods). To increase statistical power, we used functional group richness instead of species richness as explanatory variable. We fitted functional group richness either in linear (Supplementary Data 4) or log-linear (Supplementary Data 5) form. We again did not detect any significant effects of field treatment of functional group richness nor significant interactions between field treatment of functional group richness and the drought-selection history. Additionally, whether mixtures composed of plants originating from monoculture field plots or not did not affect the effects of drought-selection on biodiversity effects on productivity (Supplementary Data 3). We now report these non-significant effects as we agree that they are ecologically important (lines 203–210, 586–599).

4. A weakness of this paper is that functional traits were only measured on: 1) individuals growing in single-plant pots, or 2) individuals in mixtures before the drought (for a subset of traits - chlorophyll content, LA, LMA) and these were analyzed separately. A complete investigation of the functional basis of biodiversity

effects before, during, and after the drought would require trait measurements on individuals growing in monocultures and mixed-species pots because evolved differences plasticity in response to heterospecific vs. conspecific neighbors may underlie differences between drought and ambient selected plants. The analyses presented only examines mean trait values and so only presents part of the story. I believe this caveat should be included in the discussion at the very least.

We fully agree that a complete investigation of the functional traits of all plants growing in both monocultures and mixtures before, during and after the new drought event in the glasshouse could have provided more insights. We have supplemented this point as a limitation in the Discussion section (lines 401–424). We have also added new analyses to test the effects of drought history on multiple traits jointly (please see the response to the 54th comment of Reviewer #4).

5. Finally, while some parts of the manuscript are well written, there were many points throughout where things were not fully explained, worded confusingly, or used incorrect terminology. These instances are described in further detail in the specific comments.

We have made the clarification and revised all the relevant text according to the suggestions and comments below.

My other comments and suggestions are specific to the following parts of the manuscript:

6. L40-41: “We tested this hypothesis...” - there isn’t a hypothesis stated explicitly before this.

Revised.

7. L45-46: “... showed greater between-species complementarity” - the meaning of this is somewhat obscure because it’s not clear what exactly was measured on plants in the greenhouse.

Between-species complementarity was derived from the comparison of productivity between mixtures and monocultures. We have clarified this point (lines 43–44).

8. L48-50: I don't think the results of this study demonstrate that biodiversity causes the evolution of increased complementarity between species in response to drought. The two selection treatments compared were recurrent drought vs. ambient precipitation - while seeds of the focal species from each of these two drought treatments were collected from mixed-species communities, there were no seeds collected from drought vs. ambient precipitation in less/more diverse communities to allow the effect of biodiversity per se to be tested.

We agree that we did not demonstrate that biodiversity causes the evolution of increased complementarity. We have revised this implication to reflect our major findings that experience from past recurrent events can improve ecosystem responses to future events through transgenerational reinforcement of species complementarity (lines 48–50).

9. L60-63: Each of these ecological mechanisms could be explained more completely as they are a bit obscure as currently written. For example, the mean and variance of what? Could you please define complementarity among species – i.e. is this niche partitioning that allows them to capture resources in a complementary manner in time or space? By what mechanism does higher biodiversity lead to increased recovery?

We have deleted this sentence.

10. L68: "... via differential demographic processes" is vague and I think needs to be expanded on.

Differential demographic processes represent differential survival or proliferation of specific genotypes. We have made this clarification (lines 71–72).

11. L71 (suggestion): "... select for [trait values that confer] higher resistance..."

Corrected.

12. L75: Why does nutrient availability increase after drought?

We have deleted this sentence.

13. L79: The wording of “induce the evolution of species interactions” sounds like it is referring to the creation of new species interactions.

We have deleted this sentence.

14. L81-88: This could be generalized to: climate change (drought) may increase or decrease niche overlap between two species depending on their respective shifts in phenotypic distribution. The distinction between tolerant and intolerant genotypes and species seems arbitrary and overly complicated. For example, in Supplementary Fig. 1a, there is no reason why the phenotypic distribution of the “tolerant” species might shift right but that of the “intolerant” species stays the same, leading to reduced niche overlap. The scenarios given here in panel a and b aren’t the only possible outcomes within this framework.

We have included this suggestion on generalization (lines 86–88) and deleted Supplementary Fig. 1.

15. L88-90: I’m not sure I follow this sentence. Does this refer to plasticity in trait expression resulting from interactions with neighboring species? If this is the case, how are these changes heritable? Some clarification is necessary.

We have revised this sentence (lines 96–97). Trait selection in one species may depend on traits of other interacting species within the same communities.

16. L90-91: “Selection occurring at the community level” implies that biological communities are the units of selection. Also, is this sentence referring to selection traits in a species that is imposed by the surrounding community? Are the “individual”

species mentioned here the species that constitute the surrounding community?

We have deleted this sentence.

17. L107-108: “We harvested the biomass of every pot three times...” – after reading this and the methods it’s still not clear to me exactly what was done. Was each harvest done on one of the four plants in each pot? Or were all four plants in each pot harvested, then allowed to regrow, then harvested again etc.? It seems like the latter from the way it is written, so if this is not the case then this needs to be made much clearer.

We harvested aboveground biomass of all individuals in each pot. We have clarified this point (lines 123–124).

18. L116: Can you briefly define “biodiversity effects” here? The entire results section reports various biodiversity effects, and so a brief explanation of what positive/negative biodiversity (complementarity, sampling) effects represent would be useful.

We have added the explanations for net biodiversity effects, complementarity effects and sampling effects (lines 136–145).

19. L125-128: How did you calculate the intensities of interactions? By comparing the size of individuals in single-plant pots to those in multi-plant pots?

Yes. We have clarified the calculation of the interaction intensities (lines 149–154; Equation 2).

20. L131-134: I think the objectives should be before the methods overview so that readers understand why you are doing the things you describe there.

Agreed. We have put the objectives before the methods.

21. L133-134: “... investigate the roles of species interactions in driving the

evolutionary changes in biodiversity effects” implies that species interactions are the selective agent leading to evolutionary change, whereas the selective agent evaluated was drought vs. ambient precipitation. Perhaps “investigate the roles of species interactions in mediating the evolutionary responses to drought that underlie biodiversity effects on productivity and stability”?

Revised.

22. L143-150: Did you correct for multiple comparisons? If not, perhaps state how this compares to what would be expected just by chance.

The objective of this per-species-pair analysis was to see if particular species pairs showed significant effects of biodiversity on productivity, which corresponds to the 2nd comment raised by Reviewer #3. We did not intend this analysis to infer questions related to multiple comparisons, for example, whether there are significant differences between species pairs in their biodiversity effects on productivity. Therefore, we see no need to correct for multiple comparisons in this analysis. We have made this clarification (lines 165–167).

23. L162: Change “positively” to “positive”.

Corrected.

24. L167: Change “plant” to “plants”.

Corrected.

25. L169: To me, this result indicates that the evolutionary shifts in drought-recovery are apparent only in certain contexts (i.e., in mixed-species communities).

We have revised this sentence to indicate this context dependence.

26. L184: How is this index calculated?

We have clarified the calculation of the interaction intensities (lines 149–154 and

231–233; Equation 2).

27. L184-194: Is this paragraph only comparing individual plants to individuals in monoculture pots?

We compared interaction intensities both in monocultures and mixtures. We have made this clarification (lines 231–233).

28. L192: Change “indications” to “instances”.

Changed.

29. L195-196: How did you measure the strength of competition? Is this based on NIntM?

Yes, we calculated all the interaction or competition intensities based on NIntM. We have made this clarification (lines 231–233).

30. L227: “... was only marginally significant” – was it significant or not? The way it is written suggests that it was significant ($p < 0.05$).

The P value for this statistical test was between 0.05 and 0.10, so that we called it a marginally significant result. We have added the F and P values for this test (line 275).

31. L240, 248, 251, 267, 271, 329 etc.: “... had selected for increased niche differentiation”, “... selection for more positive biodiversity effects”, “... selection for increased complementarity”, “selection for niche differentiation” etc. – this is bit of a bugbear for me, and here’s why... To say that there was selection for something (e.g., increased niche differentiation, more positive biodiversity effects) means that that was the mechanism by which it increased fitness of individuals with those traits and so was favored by natural selection. While increased niche differentiation, for example, might be an effect of evolutionary responses to drought, it is not necessarily the cause of the evolutionary response (i.e., the mechanism by which individuals had higher

fitness). We don't know the mechanisms by which some individuals in drought plots had higher fitness; perhaps it was greater niche differentiation or perhaps it was something else, but this was never tested. To illustrate with an example, selection might favor greater rooting depth under drought because it plants to access deeper water (i.e., selection for a specific property that enhances fitness under drought – being more drought tolerant), which could then result in plants that differ more from other species in their rooting depth (i.e., selection of an object – plants with lower niche overlap with heterospecifics). The mechanism by which greater rooting depth increased fitness is because it allowed plants to exploit deeper water, the increase in niche overlap/biodiversity effects are just side effects. It is important to distinguish between the function of an adaptation (selection for) from the effect it has (selection of).

We have revised all the relevant texts to better distinguish between the function of an adaptation from the effect it has.

32. L268-270: This explanation of how drought can result in an increase in nutrient availability should be in the introduction where the idea is first mentioned.

We have restructured the introduction. Now we only introduce this idea in the discussion.

33. L271: It's not immediately obvious how a large flux of resources increases the potential for niche partitioning. Could you cite a study that shows this effect?

A release of constrained resources after drought may shift competition for a single resource (water) during the most stressful phase of drought to multiple resources after drought, and thus increase the potential for niche partitioning after drought. We have added this explanation (lines 320–323).

34. L272: Why specifically epigenetic variation? Epigenetic variation is only one source of non-genetic variation.

We have revised the texts to include other sources of non-genetic variation, such as non-heritable parental effect (lines 378–379).

35. L273: Explaining which results?

“The results” represent the transgenerational reinforcement of species complementarity. We have clarified this point (lines 379–381).

36. L273-276: What do you mean by adaptive changes? What was measured? And what was the selection regime? Are those seeds the same as the seeds used in this experiment?

We have added the detailed explanations about the measures, selection regime and the seed sources in this experiment (lines 381–387).

37. L286: Again, “induces the evolution of species interactions” sounds like the creation of new interactions.

We have revised it as “induces changes in species interactions”.

38. L276: I think discussion of trait results should be in its own paragraph.

Done.

39. L276-292: What about drought-induced evolution of plasticity in traits in response to the presence of neighboring plants? Could it be that measuring traits on single individuals (i.e., mean trait values) don’t capture the full evolutionary consequences of drought?

We have included this helpful suggestion (lines 409–413).

40. L299-300: “... biodiversity-dependent evolutionary adaptation...” – this study never tested how different levels biodiversity influenced evolutionary adaptation as the study compared plants from drought vs. ambient plots. Perhaps “evolutionary

responses to drought, the expression of which depend on levels of biodiversity” or something like that?

Revised.

41. L320-326: This paragraph seems out of place and is not connected to the findings of this study.

We have deleted this paragraph.

42. L323: What do you mean by “a big picture”?

We have deleted this paragraph.

43. L336-339: This should be in the results.

We keep this information in the discussion, because we cannot find a good place to fill it into the results. It is a minor result and does not fit well in the current framework of the results.

44. L372: Seeds were collected from which mixtures? 60 species mixtures?

We have added a table about the information of seed sources in Supplementary Data 2. Seeds of each species of each drought treatment history were collected from multiple plots (interquartile range: [4.75, 9.00]) in Jena, in which the functional group richness ranged from 1–4.

45. L377: What defines a “small” and “tall” herb?

Small and tall herbs are primarily differentiated by their canopy heights of vegetative and flowering plants. We have added the reference (Roscher et al. 2004) containing the detailed explanations about the classification of the functional groups (lines 476–477).

46. L379: “... planted seedings...” contradicts the later sentence “Seeds... were

sown”.

We germinated the seeds in petri dishes and then transplanted the seedlings into pots.

We have made this clarification (line 484).

47. L385: What do you mean by “perfect matches in seed families”?

We have deleted this sentence.

48. L403: Did you harvest a single plant from each pot, or did you harvest the aboveground biomass of all the plants in a pot?

We harvested aboveground biomass of all individuals in each pot. We have made this clarification (line 506).

49. L417: What is the additive partitioning approach? Please explain this!

We have added the equation (Equation 1) and explanations for the additive partitioning approach (lines 534–548).

50. L420-422: “For the monocultures...” – I found this whole sentence to be quite confusing.

Revised.

51. L429: What is the response variable in these models? NEs, CEs, and SEs?

Yes. We have made the clarification (line 560).

52. L447: Why did you also run these separate treatment models?

We ran these separate treatment models to test the significance of biodiversity effects (NEs, CEs and SEs) on productivity for each drought treatment history and harvest.

We have made the clarification (lines 577–579).

53. L455-459: This is confusing, as earlier in the methods on L381-382 it is stated

that “Seeds from drought and ambient plots of the same large plot in Jena were sown...”. This suggests that seeds weren’t all collected from the same large plot, which I would assume have the same functional richness (as this is a plot-level property).

Seeds of each species of each drought treatment history were collected from 2–10 (interquartile range: [4.75, 9.00]) large plots in Jena, in which the functional group richness ranged from 1–4 (Supplementary Data 2). Different large plots in Jena can have the same or different functional group richness. We have clarified this point (lines 588–594). We believe that this comment is related to the 44th and 47th comments of Reviewer #4; please also see the responses to these comments.

54. L558-565: With so many traits, it seems like it would be appropriate to use MANOVA first.

We used two alternative approaches to explore the multiple traits jointly. (1) We performed a principal component analysis with all traits and retained the first two principal axes (PC1 and PC2), which accounted for 39.06% and 22.3% of the total variation, respectively. In this way, we reduced the five correlated dimensions (corresponding to the five traits measured before the new drought treatment) of traits into two orthogonal axes. Then, we used PC1 and PC2 as response variables in mixed-effect models, separately. We fitted the models with the same fixed- and random-effects terms as those using each separate trait as the response variable. We did not detect significant effect of selection treatment history for neither PC1 ($F_{1,5.7} = 1.206, P = 0.316$) nor PC2 ($F_{1,6.1} = 0.005, P = 0.945$). (2) We pooled the five traits as a single response variable in a mixed-effect model (corresponding to multivariate analysis of variance). Block, trait group (a factor with five levels), treatment history and the interaction between trait group and treatment history were set as fixed-effects terms; species and its interactions with trait group and treatment history and their three-way interaction were set as random effects. We did not detect significant effects of selection treatment ($F_{1,6.0} = 2.183, P = 0.190$) nor its interaction with trait group

($F_{4,34.4} = 0.217, P = 0.927$). These two new analyses yielded the same conclusion as those based on single traits. That is, selection treatment history did not cause any significant difference in trait values before the new drought treatment. We have added information associated with these two new analyses (lines 704–720).

55. L570-572: How did you standardize trait values before calculating Euclidean distance?

We standardized each trait to mean zero and unit standard deviation. We have clarified this point (line 728).

56. Figures 2 and 3: Figures 2 and 3 more or less show the same information in different ways. I wonder if Fig.2 should be in the supplement rather than the main article as it seems slightly redundant.

Figures 5 and 6: As for Figs. 2 and 3, I wonder if Figure 5 is better in the supplementary info as it displays much of the same information as in Fig. 6.

We have moved Figs. 2 into the supplementary information, and deleted Fig. 5 to simplify the presentation.

57. Supplementary Table 1: Typo in “2-speices”

Corrected.

REVIEWER COMMENTS

Reviewer #1 (Remarks to the Author):

I appreciate the efforts of the authors to address my comments. In general, they did a great work. I appreciate that they generally replaced the „evolutionary process” to „transgenerational effect”, and provide a balanced view on what processes may cause the observed pattern. They also added many more details on different parts of the methodology.

To me the short time between seeding and fully-developed status (14 weeks for perennial species!), as well as the repeated full-cuts of aboveground biomass of the same individuals sound strange, but the authors provided arguments for why these things could work in their experimental system.

There is one thing I would suggest for consideration. I know that „complementarity effect” is a well-established term, but it is probably used only within a subset of ecologists, and I think that many ecologists may not exactly know what it exactly is when reading it (including myself). Therefore, I think it would be useful to define it early in the manuscript (Introduction, or already in the Abstract)., In addition, I am not sure it should be put into the title, What about using „biodiversity effect” in the title instead of „species complementarity”; the authors already use this in some parts of the discussion. I know that it is not a true synonym, but it would probably be much more appealing in the title (would probably attract more attention). I am not pushing this; this is just a thought for consideration.

Reviewer #3 (Remarks to the Author):

The authors have undertaken revision of the paper reviewed in 2019. The revision and response to reviewer’s comments were thorough and thoughtful, overall. I appreciated the attention to detail and honesty in all of the responses, as well as all caveats in the new manuscript text. The results showed that droughted plants in 2-species mixtures recover faster after a subsequent drought, however, less than half of the species pairs showed this complementarity response, there were no patterns of which species pairs and there were no apparent differences in measured traits. I think these patterns are interesting, potentially very important and while the authors suggest multiple possible mechanisms, there is no explanation or mechanism for these patterns. This leaves the reader unsatisfied and leads to doubt about the methods (and thus interpretation). See additional details below.

Line 1: Place “drought memory” in quotes since there is no mechanism for this memory?

Line 1: replace “new” with “recurrent or subsequent” drought (in title and throughout text)?

Line 122: This is the start of a section on the cutting methods – the explanation for cutting biomass three times in a growing season was explained in the response to reviewers but not here. It is still not clear why there could be compensatory growth responses to cutting.

Line 134: Why aren’t these analyses directional hypotheses? The hypothesis in the Intro is very general

but analyses of the data are addressing specific hypotheses. I think this would help with readability if stated as explicit hypotheses.

Lines 157-161: Omit, results just before results section

Lines 164: section title 'Effects of drought-selection on biodiversity effects on productivity' too wordy and hard to follow, perhaps instead 'Effects of drought on complementarity'?

Line 201-202: unclear what is meant here – explain

Lines 234-235: I thought all pots have 4 individuals (i.e., equal density) – do you mean monoculture vs. mixture?

Line 381-383: The result of the previously published paper has no bearing on these results or interpretation of this result. This is a legacy of the previous version that was focused on selection, a concept that has been broadened

Line 44: they “may” enhance

Reviewer #4 (Remarks to the Author):

The authors of “Drought memory increases species complementarity in response to a new drought in experimental plant communities” have substantially revised and improved the original manuscript, and it was a pleasure to read this revised version. The authors have largely addressed the concerns I raised about the initial submission.

My comments, suggestions, and questions are relatively minor and specific to the following parts of the manuscript.

Title (and throughout): Although I understand the reasoning behind the change in title, I find the term “drought memory” a bit New Age and might be confusing to potential readers. I think perhaps something like “Long-term drought exposure increases species complementarity of offspring in response to subsequent drought in experimental plant communities” or something along those lines explains a bit more direct.

L38 (and throughout): Similarly, I find the terms “experience” and “past events” are a bit vague/odd, I prefer “exposure to extreme climatic events” or something like that.

L40 (and throughout): I understand why they authors have chosen to use “transgenerational effects” as a broad umbrella term for what could be genetic evolution, transgenerational plasticity, epigenetic inheritance etc., but I think the term “transgenerational” tends to be strongly associated with plasticity and so I wonder if for clarity it could be switched to something else such as “inclusive inherited effects” – to borrow terminology from the emerging extended evolutionary synthesis.

L119 (suggestion): “... in the ambient treatment”.

L143 (suggestion): “... that are more productive in monocultures.”

L202: Supplementary Table 3 doesn't support the statement that effects were mostly driven by positive

sampling effects because it only shows the p-values for each biodiversity effect as opposed to quantifying the amount they contributed (i.e., their effect size).

L216 (suggestion): Delete “in mixtures”.

L306: I was under the impression that the SGH doesn't only propose that facilitation will be more common under stress, but also that the relative importance of competition for reducing growth will be reduced. In other words, equally frequent competition that was less important in each case in drought (with same frequency of facilitation) is still consistent with the SGH.

L316 (and throughout): I previously raised the point of distinguishing between “selection of” and “selection for” and appreciate the authors addressing this in the revision. However, given that we do not know that the observed effects are due to genetic evolution, transgenerational plasticity etc., I wonder if it would be more appropriate to simply say “Increased niche differentiation in drought-selected...” rather than “The selection of niche differentiation...”.

L349 (suggestion): I think “formally test” is more appropriate than “statistically confirm”.

Discussion: Overall I found the discussion to be a bit of a slog. It is well written and nicely structured, but very long and so I encourage the authors to really focus on the key findings and implications of the study and connect the readers to the “bigger picture” so to speak.

L471: Collected seeds from as low as 4 plants across the whole experiment seems very low and I would expect this could lead to large effects of sampling error. Is this a minimum of 4 per treatment per species or 4 per species across treatments. How many species were represented by similarly low numbers of maternal plants?

L482: Beginning of the drought treatment was 8 years before. Any ideas about the densities of individuals at the time of seed collection?

L498: “Species pairs...” - I think there is a typo or missing word in this sentence as it doesn't really make sense.

Reviewer #1 (Remarks to the Author):

I appreciate the efforts of the authors to address my comments. In general, they did a great work. I appreciate that they generally replaced the „evolutionary process” to „transgenerational effect”, and provide a balanced view on what processes may cause the observed pattern. They also added many more details on different parts of the methodology. To me the short time between seeding and fully-developed status (14 weeks for perennial species!), as well as the repeated full-cuts of aboveground biomass of the same individuals sound strange, but the authors provided arguments for why these things could work in their experimental system.

There is one thing I would suggest for consideration. I know that „complementarity effect” is a well-established term, but it is probably used only within a subset of ecologist, and I think that many ecologist may not exactly know what it exactly is when reading it (including myself). Therefore, I think it would be useful to define it early in the manuscript (Introduction, or already in the Abstract)., In addition, I am not sure it should be put into the title, What about using „biodiversity effect” in the title instead of „species complementarity”; the authors already use this in some parts of the discussion. I know that it is not a true synonym, but it would probably be much more appealing in the title (would probably attract more attention). I am not pushing this; this is just a thought for consideration.

Thank you for the helpful comments. We have added the explanation for the repeated cutting in the Introduction section (lines 132–134). We have also clarified the meanings of species complementarity early in the Introduction section (lines 60–63). We did not replace “species complementarity” with “biodiversity effect” in the title, because “biodiversity effect” is so general a term that it may cause confusion. For example, it could refer to the effects of biodiversity on productivity or stability in this paper.

Reviewer #3 (Remarks to the Author):

The authors have undertaken revision of the paper reviewed in 2019. The revision and response to reviewer's comments were thorough and thoughtful, overall. I appreciated the attention to detail and honesty in all of the responses, as well as all caveats in the new manuscript text. The results showed that droughted plants in 2-species mixtures recover faster after a subsequent drought, however, less than half of the species pairs showed this complementarity response, there were no patterns of which species pairs and there were no apparent differences in measured traits. I think these patterns are interesting, potentially very important and while the authors suggest multiple possible mechanisms, there is no explanation or mechanism for these patterns. This leaves the reader unsatisfied and leads to doubt about the methods (and thus interpretation). See additional details below.

Thank you for the helpful comments. We apologize that we were unclear about the results of per species-pair tests. We understand that the confusion arose from the presentation of both, (1) results from the separate analysis of species pairs and (2) results from the combined statistical analysis. Whereas for (1) 11 out of 12 species pairs of drought-selected plants showed increased complementarity effects (CEs) in response to the subsequent drought, five of the increase were significant. However, with (2) we showed that the increase is a general and significant effect when tested against the variation in individual effects among species pairs (Fig. 2; Table 1). This variation among species pairs did not show any specific patterns as we showed in Supplementary Data 1. The non-significances in the seven individual tests (1) are not in the contrast to the significance of the general test (2) because statistical power was of course lower for the individual tests, and thus significance was not reached in seven cases, six of which nevertheless were in the direction of the general test, to which they contributed in (2). We have reformulated the corresponding paragraphs about the individual and combined analyses in the Results section (lines 165–195). To avoid confusion, we have deleted the sentence counting the numbers of species pairs with significant individual tests.

Line 1: Place “drought memory” in quotes since there is no mechanism for this memory?

We have replaced “drought memory” with “drought-exposure history”.

Line 1: replace “new” with “recurrent or subsequent” drought (in title and throughout text)?

We have replaced “new” with “subsequent” in the title and throughout the text.

Line 122: This is the start of a section on the cutting methods – the explanation for cutting biomass three times in a growing season was explained in the response to reviewers but not here. It is still not clear why there could be compensatory growth responses to cutting.

We have added the explanation for the repeated cutting (lines 132–134): “This harvesting procedure mimics the common cutting management of the species in the field (and in comparable grasslands in the region), where up to four harvests per growing season are being made.”

Line 134: Why aren’t these analyses directional hypotheses? The hypothesis in the Intro is very general but analyses of the data are addressing specific hypotheses. I think this would help with readability if stated as explicit hypotheses.

We have clarified the linkages between these analyses and the proposed research questions and hypotheses to improve the readability (lines 104–116, 138–139). We kept the two-sided significance tests because they are more conservative and because we could not be sure in advance that effects would be unidirectional.

Lines 157-161: Omit, results just before results section

Deleted.

Lines 164: section title ‘Effects of drought-selection on biodiversity effects on productivity’ too wordy and hard to follow, perhaps instead ‘Effects of drought on complementarity’?

We have simplified all the sub-section titles in the Results section.

Line 201-202: unclear what is meant here – explain

We have revised this sentence (lines 193–195): “Before the subsequent drought event, the positive biodiversity effects were mainly due to positive sampling effects (Fig. 2a, d, g).”

Lines 234-235: I thought all pots have 4 individuals (i.e., equal density) – do you mean monoculture vs. mixture?

We grew plants individually (individual-plant pots), in monocultures (four-plant pots) or mixtures (four-plant pots). We measured plant interaction intensity by comparing plant biomass between individual-plant pots vs. four-plant pots. We have revised this sentence to clarify this point (lines 226–228): “The interaction intensity was mostly negative (Supplementary Fig. 4), indicating that plants in pots with four individuals (monocultures or mixtures) had less biomass than plants in pots with one individual.”

Line 381-383: The result of the previously published paper has no bearing on these results or interpretation of this result. This is a legacy of the previous version that was focused on selection, a concept that has been broadened

We discussed the results of this published paper to explain that both genetic and non-genetic processes were possible in driving the observed transgenerational effects. This cited paper corresponds to the genetic processes. We also cited and discussed other papers on non-genetic processes in the same paragraph. We have revised this paragraph and made this clarification (lines 349–357).

Line 44: they “may” enhance

Revised

Reviewer #4 (Remarks to the Author):

The authors of “Drought memory increases species complementarity in response to a new drought in experimental plant communities” have substantially revised and improved the original manuscript, and it was a pleasure to read this revised version. The authors have largely addressed the concerns I raised about the initial submission.

My comments, suggestions, and questions are relatively minor and specific to the following parts of the manuscript.

Title (and throughout): Although I understand the reasoning behind the change in title, I find the term “drought memory” a bit New Age and might be confusing to potential readers. I think perhaps something like “Long-term drought exposure increases species complementarity of offspring in response to subsequent drought in experimental plant communities” or something along those lines explains is a bit more direct.

We have revised the title to “Drought-exposure history increases complementarity between plant species in response to a subsequent drought”.

L38 (and throughout): Similarly, I find the terms “experience” and “past events” are a bit vague/odd, I prefer “exposure to extreme climatic events” or something like that.

Revised.

L40 (and throughout): I understand why they authors have chosen to use “transgenerational effects” as a broad umbrella term for what could be genetic evolution, transgenerational plasticity, epigenetic inheritance etc., but I think the term “transgenerational” tends to be strongly associated with plasticity and so I wonder if for clarity it could be switched to something else such as “inclusive inherited effects” – to borrow terminology from the emerging extended evolutionary synthesis.

We have added this alternative terminology “inclusive inheritance” early in the Introduction section and clarified that transgenerational effects or inclusive inheritance can arise from both genetic and non-genetic effects (lines 68–69).

L119 (suggestion): "... in the ambient treatment".

Revised

L143 (suggestion): "... that are more productive in monocultures."

Revised

L202: Supplementary Table 3 doesn't support the statement that effects were mostly driven by positive sampling effects because it only shows the p-values for each biodiversity effect as opposed to quantifying the amount they contributed (i.e., their effect size).

We have deleted the citation of Supplementary Table 3 here.

L216 (suggestion): Delete "in mixtures".

Deleted

L306: I was under the impression that the SGH doesn't only propose that facilitation will be more common under stress, but also that the relative importance of competition for reducing growth will be reduced. In other words, equally frequent competition that was less important in each case in drought (with same frequency of facilitation) is still consistent with the SGH. You are right that SGH refers to both increased facilitation and reduced competition under stressful conditions (the way we refer to it at lines 110–112). We have deleted the reference to SGH and the corresponding text from the above text (lines 298–301).

L316 (and throughout): I previously raised the point of distinguishing between "selection of" and "selection for" and appreciate the authors addressing this in the revision. However, given that we do not know that the observed effects are due to genetic evolution, transgenerational plasticity etc., I wonder if it would be more appropriate to simply say "Increased niche differentiation in drought-selected..." rather than "The selection of niche differentiation...".

Revised.

L349 (suggestion): I think “formally test” is more appropriate than “statistically confirm”.

Revised

Discussion: Overall I found the discussion to be a bit of a slog. It is well written and nicely structured, but very long and so I encourage the authors to really focus on the key findings and implications of the study and connect the readers to the “bigger picture” so to speak.

We have simplified the discussion as suggested.

L471: Collected seeds from as low as 4 plants across the whole experiment seems very low and I would expect this could lead to large effects of sampling error. Is this a minimum of 4 per treatment per species or 4 per species across treatments. How many species were represented by similarly low numbers of maternal plants?

Collected seeds were from at least 4 plants per selection treatment per species, and at least 8 plants per species across treatments. Ten of the 12 species had their seeds collected from > 4 plants per selection treatment. We have clarified this point at lines 435–436 and included the seed collection information in Supplementary Data 2.

L482: Beginning of the drought treatment was 8 years before. Any ideas about the densities of individuals at the time of seed collection?

Due to the perennial growth of the species, we unfortunately do not have the data about the densities of genetic individuals at the time of seed collection. But we know that all the species collected for the glasshouse experiment were well represented in the Jena plots at the time of seed collection.

L498: “Species pairs...” - I think there is a typo or missing word in this sentence as it doesn't really make sense.

We have reworded this sentence. Now it appears as “Species pairs composed of *Crepis biennis* or *Lotus corniculatus* had low numbers of replicates.”

REVIEWERS' COMMENTS

Reviewer #1 (Remarks to the Author):

I had only minor comments in the previous round of review, and I am now satisfied with the way the authors addressed my suggestions.

Reviewer #4 (Remarks to the Author):

The authors of “Drought memory increases species complementarity in response to a new drought in experimental plant communities” have revised and improved the revised manuscript, and I have no further comments or suggestions. I would like to congratulate them for their hard work.

Reviewer #1 (Remarks to the Author):

I had only minor comments in the previous round of review, and I am now satisfied with the way the authors addressed my suggestions.

Response: Thanks for your time and constructive suggestions. The reviewer did not have further comments so that we did not make any corresponding revision.

Reviewer #4 (Remarks to the Author):

The authors of “Drought memory increases species complementarity in response to a new drought in experimental plant communities” have revised and improved the revised manuscript, and I have no further comments or suggestions. I would like to congratulate them for their hard work.

Response: Thanks for your time and constructive suggestions. The reviewer did not have further comments so that we did not make any corresponding revision.